# Unveiling the Therapeutic Potential of Folate-Dependent One-Carbon Metabolism in Cancer and Neurodegeneration

**DOI:** 10.3390/ijms25179339

**Published:** 2024-08-28

**Authors:** Ana Filipa Sobral, Andrea Cunha, Vera Silva, Eva Gil-Martins, Renata Silva, Daniel José Barbosa

**Affiliations:** 1Associate Laboratory i4HB—Institute for Health and Bioeconomy, University Institute of Health Sciences—CESPU, 4585-116 Gandra, Portugal; ana.sobral@iucs.cespu.pt; 2UCIBIO—Applied Molecular Biosciences Unit, Toxicologic Pathology Research Laboratory, University Institute of Health Sciences (1H-TOXRUN, IUCS-CESPU), 4585-116 Gandra, Portugal; 3UNIPRO—Oral Pathology and Rehabilitation Research Unit, University Institute of Health Sciences—CESPU, 4585-116 Gandra, Portugal; andrea.cunha@iucs.cespu.pt; 4Associate Laboratory i4HB—Institute for Health and Bioeconomy, Faculty of Pharmacy, University of Porto, 4050-313 Porto, Portugal; veralssilva17@gmail.com (V.S.); evagilmartins18@gmail.com (E.G.-M.); rsilva@ff.up.pt (R.S.); 5UCIBIO—Applied Molecular Biosciences Unit, Laboratory of Toxicology, Department of Biological Sciences, Faculty of Pharmacy, University of Porto, 4050-313 Porto, Portugal; 6CIQUP-IMS/Department of Chemistry and Biochemistry, Faculty of Sciences, University of Porto, 4169-007 Porto, Portugal; 7UCIBIO—Applied Molecular Biosciences Unit, Translational Toxicology Research Laboratory, University Institute of Health Sciences (1H-TOXRUN, IUCS-CESPU), 4585-116 Gandra, Portugal

**Keywords:** folate, folate-dependent one-carbon metabolism, folate cycle, methionine cycle, cancer, neurodegeneration, folate-targeted therapies

## Abstract

Cellular metabolism is crucial for various physiological processes, with folate-dependent one-carbon (1C) metabolism playing a pivotal role. Folate, a B vitamin, is a key cofactor in this pathway, supporting DNA synthesis, methylation processes, and antioxidant defenses. In dividing cells, folate facilitates nucleotide biosynthesis, ensuring genomic stability and preventing carcinogenesis. Additionally, in neurodevelopment, folate is essential for neural tube closure and central nervous system formation. Thus, dysregulation of folate metabolism can contribute to pathologies such as cancer, severe birth defects, and neurodegenerative diseases. Epidemiological evidence highlights folate’s impact on disease risk and its potential as a therapeutic target. In cancer, antifolate drugs that inhibit key enzymes of folate-dependent 1C metabolism and strategies targeting folate receptors are current therapeutic options. However, folate’s impact on cancer risk is complex, varying among cancer types and dietary contexts. In neurodegenerative conditions, including Alzheimer’s and Parkinson’s diseases, folate deficiency exacerbates cognitive decline through elevated homocysteine levels, contributing to neuronal damage. Clinical trials of folic acid supplementation show mixed outcomes, underscoring the complexities of its neuroprotective effects. This review integrates current knowledge on folate metabolism in cancer and neurodegeneration, exploring molecular mechanisms, clinical implications, and therapeutic strategies, which can provide crucial information for advancing treatments.

## 1. Introduction

The complex and interconnected pathways of cellular metabolism are responsible for fundamental processes necessary to support the function of living organisms [1]. Among these pathways, folate-dependent one-carbon (1C) metabolism represents a key player, as it is involved in a multitude of essential biochemical reactions critical for cellular homeostasis and function [2,3].

Folate, a water-soluble vitamin of the B complex (B9), works as a crucial cofactor in 1C metabolism, as it is involved in the transfer of 1C units for biosynthetic reactions essential for cell growth and proliferation. Central to these processes are the synthesis of purines (adenine and guanine) and thymidylate (an intermediate for de novo synthesis of thymidine), which are critical components of DNA replication and repair, and the regulation of amino acid homeostasis [2,3]. Folate also plays a fundamental role in providing methyl groups for DNA methylation, a key epigenetic modification that regulates gene expression and chromatin structure [3,4,5]. Therefore, dysregulation of folate metabolism can lead to abnormal DNA methylation, disrupting normal cellular function and contributing to disease pathogenesis [6,7]. Particularly in cancer, changes in DNA methylation have been linked to the silencing of tumor suppressor genes and activation of oncogenes, favoring tumorigenesis and metastasis [8,9,10].

Beyond its role in nucleotide synthesis and DNA methylation, folate metabolism also regulates cellular redox balance and antioxidant defense mechanisms [11]. It regulates the synthesis of glutathione, a major cellular antioxidant, crucial for maintaining redox homeostasis and protecting cells from oxidative stress events [12,13]. Therefore, the dysregulation of folate-mediated redox pathways has been implicated not only in the pathogenesis of cancer [14,15,16,17] but also in neurodevelopmental and mental disorders [18,19,20], where increased oxidative stress and impaired antioxidant defenses contribute to disease pathogenesis and progression.

The complex interplay between folate metabolism and disease pathogenesis is further underscored by the association between folate levels and disease risk. Epidemiological studies have consistently reported an inverse relationship between dietary folate intake and the incidence of certain cancers, particularly colorectal cancer [21,22]. Similarly, folate deficiency has been associated with an increased risk of neurodegenerative disorders, where folate supplementation improves cognitive function [23,24,25]. The interplay among folate metabolism, neurotransmitter synthesis, and oxidative stress [26] highlights the complex relationship among the pathophysiological mechanisms underlying neurodegeneration. These observations emphasize the potential significance of modulating folate metabolism as a preventive or therapeutic strategy for cancer and neurodegeneration.

In line with this idea, over the last years, significant efforts have been made to elucidate the molecular mechanisms underlying the dysregulation of folate metabolism in disease and explore the therapeutic potential of targeting this metabolic pathway. Preclinical studies have identified promising pharmacological agents that modulate folate metabolism and have demonstrated efficacy in inhibiting tumor growth and neurodegeneration [27,28]. Therefore, this metabolic pathway may unveil tremendous potential for targeted therapy in prevalent human diseases.

In this manuscript, we provide a comprehensive overview of the current knowledge of folate-dependent 1C metabolism in cancer and neurodegeneration. This includes the molecular mechanisms underlying its dysregulation, the clinical implications of altered folate metabolism in cancer and neurodegeneration, and putative therapeutic strategies targeting this metabolic pathway and their benefits in such diseases.

## 2. Folate-Dependent One-Carbon Metabolism: An Overview

Folate-dependent 1C metabolism refers to a complex network of biochemical reactions that involves the transfer of one-carbon units to various substrates (Figure 1). It occurs in the cytoplasm, nucleus, and mitochondria, is differentially regulated within these compartments, and plays a crucial role in numerous processes essential for cell function and survival [29]. These processes include the biosynthesis of essential molecules (nucleotides, polyamines, proteins, phospholipids, and creatine) and energy production. In addition, 1C metabolism also supports processes such as redox homeostasis, amino acid metabolism, histone modification, and the epigenetic regulation of DNA and ribonucleic acid (RNA) methylation [29,30]. The primary regulators of 1C metabolism, which are all essential dietary requirements, include methionine, folate (B9), and vitamin B12 [31].

Folate-dependent 1C metabolism includes two main cycles, the folate cycle and the methionine cycle (Figure 1). The folate cycle is considered the core of this broader set of processes that integrate 1C metabolism. Specifically, through an intercompartmental network of interlinked reactions, the folate cycle supports the de novo synthesis of purines and thymidylate and provides the intermediate 5-methyltetrahydrofolate (5-MTHF), which ensures the transfer of 1C units for methylation reactions in the methionine cycle [28]. The chemical structures of folic acid, folate (in both polyglutamate and monoglutamate forms), and various folate derivatives that participate in 1C metabolism are depicted in Figure 2.

Upon entering the cell, and before it can enter the folate cycle, folic acid is sequentially reduced first to dihydrofolate (DHF) and then to tetrahydrofolate (THF) by dihydrofolate reductase (DHFR), using reduced nicotinamide adenine dinucleotide phosphate (NADPH) as a cofactor [33]. Thereafter, folate metabolism operates through two parallel and complementary pathways located in the cytosol and mitochondria [28]. The first reaction is catalyzed by serine hydroxymethyltransferase (SHMT), specifically SHMT1 (in the cytoplasm) and SHMT2 (in the mitochondria). In this step, serine is converted to glycine, and its 1C unit is transferred to THF, resulting in the formation of 5,10-methylenetetrahydrofolate (5,10-MTHF). In the cytosol, 5,10-MTHF participates in two different processes as follows: (1) in thymidylate synthesis, where 5,10-MTHF functions as a methyl donor for the conversion of deoxyuridine-5′-monophosphate (dUMP) into deoxythymidine-5′-monophosphate (dTMP; thymidylate), catalyzed by thymidylate synthase (TYMS), recycling DHF to continue the cycle and (2) to support the methionine cycle, where, in a reaction catalyzed by methylenetetrahydrofolate reductase (MTHFR), a NADPH-dependent enzyme using vitamin B2 as a cofactor, 5,10-MTHF is reduced to 5-MTHF. By donating a methyl group in the methionine cycle, 5-MTHF regenerates methionine from homocysteine (HCY) in a reaction catalyzed by methionine synthase (MTR) using cobalamin (vitamin B12) as a cofactor [28,31]. Subsequently, 5,10-MTHF is converted to 10-formyl-tetrahydrofolate (10-formyl-THF), the most oxidized form of folate carbon. This reaction, when it occurs in the cytosol, is catalyzed by methylenetetrahydrofolate dehydrogenase 1 (MTHFD1), a trifunctional enzyme with 10-formyl-THF synthase, 5,10-MTHF dehydrogenase, and 5,10-MTHF cyclohydrolase activities, with reduction of the nicotinamide adenine dinucleotide phosphate oxidized form (NADP^+^). On the other hand, when it occurs in the mitochondria, the conversion of 5,10-MTHF into 10-formyl-THF is catalyzed by methylenetetrahydrofolate dehydrogenase 2 (MTHFD2) or methylenetetrahydrofolate dehydrogenase 2-like (MTHFD2L), which exhibit bifunctional cyclohydrolase and dehydrogenase activities, with NADP^+^ reduction to NADPH. The newly formed cytosolic 10-formyl-THF is crucial for de novo purine synthesis, while mitochondrial 10-formyl-THF, by producing formyl-methionine transfer ribonucleic acid (tRNA), is used to initiate mitochondrial protein translation. Furthermore, because of the inability of 10-formyl-THF to pass across the mitochondrial membranes, it needs to be hydrolyzed to formate in a reaction catalyzed by the methylenetetrahydrofolate dehydrogenase 1-like (MTHFD1L) enzyme while also phosphorylating adenosine-5′-diphosphate (ADP) to adenosine-5′-triphosphate (ATP). In the cytosol, 10-formyl-THF can also be converted to formate by MTHFD1, with ADP phosphorylation. Finally, through a reaction catalyzed by cytosolic 10-formyltetrahydrofolate dehydrogenase (ALDH1L1) or mitochondrial 10-formyltetrahydrofolate dehydrogenase (ALDH1L2), 10-formyl-THF is completely oxidized to CO_2_ and eliminated [28] (Figure 1).

Closely interconnected with the folate cycle is the methionine cycle, an interaction crucial for maintaining amino acid homeostasis and supporting several cellular functions [33]. The methionine cycle starts with the transfer of an adenosyl group to methionine (imported from the extracellular space) in a reaction catalyzed by methionine adenosyltransferase 2A (MAT2A), which consumes ATP. This produces S-adenosylmethionine (SAM), a crucial molecule in the biosynthesis of polyamines such as putrescine, spermidine, and spermine. SAM also serves as a methyl donor for multiple methylation reactions catalyzed by methyltransferases (MTs) specific for RNA, DNA, histones, and proteins [31]. During methylation reactions, SAM is demethylated to S-adenosylhomocysteine (SAH), which is then hydrolyzed to HCY, in a reaction catalyzed by S-adenosylhomocysteine hydrolase (AHCY). HCY, as previously mentioned, can be converted into methionine by receiving a methyl group from 5-MTHF, or it can enter the transsulfuration pathway, where it is converted into cysteine. Cysteine then contributes to the biosynthesis of reduced glutathione (GSH), important for the redox homeostasis of cells [31] (Figure 1).

In summary, 1C metabolism is a complex and essential network of biochemical reactions critical for various cellular functions, including nucleotide biosynthesis, energy production, redox homeostasis, and epigenetic regulation (DNA and histone methylation). This metabolic pathway relies on key dietary nutrients such as methionine, folate, and vitamin B12 and operates across different cellular compartments with distinct regulatory mechanisms. Thus, a disruption of 1C metabolism can lead to various health issues, including developmental disorders and certain types of cancer. Understanding this metabolic pathway not only enhances our knowledge on the fundamental biological processes but also opens avenues for potential therapeutic strategies aimed at addressing metabolic-related diseases.

## 3. Sources of Dietary Folate and Its Bioavailability

Folate is required for several biological processes in the human body, such as DNA synthesis and red blood cell formation. Considering that eukaryotic cells do not have the intrinsic capacity to produce folate, because of the lack of enzymes necessary for its de novo synthesis [34], humans must obtain folate from their diet or through supplementation to prevent health problems [35] (for details, see Section 5, Folate Metabolism and Cancer, and Section 6, Folate Metabolism and Neurodegeneration). Adequate folate intake is particularly essential during periods of rapid growth and development, such as pregnancy and infancy [36].

Folate exists in different forms, including dietary folate (naturally present in foods) and synthetic folic acid found in fortified foods and supplements. Natural food sources of folate include leafy green vegetables, legumes, fruits, and whole grains [37]. Table 1 lists the richest natural food sources of folate and their nutrient content.

Leafy green vegetables, including spinach, kale, collard greens, and Swiss chard, are the richest sources of folate, representing good choices to be included in any diet [38]. For example, one cup of cooked spinach (125 g) contains about 60% of the recommended daily intake of folate for adults [39]. Therefore, the addition of leafy green vegetables to salads, soups, or drinks represents a good alternative for adequate import of folate from diet.

Legumes, including beans, lentils, chickpeas, and peas, represent another dietary group enriched in folate [38]. For example, one cup of lentils contains about 99% of the recommended daily intake of folate for adults [40]. Therefore, the addition of legumes to salads, or as a main dish, not only enhances the nutritional profile of meals, as also represents a good alternative for guaranteeing adequate folate intake.

**Table 1 ijms-25-09339-t001:** The richest natural food sources of folate and their nutrient contents.

Natural Food Sources of Folate		Folate Content (µg/100 g)	% of Recommended Daily Intake (per 100 g) ^1^	Reference
**Leafy green** **vegetables**	Spinach	194	48.5	[41]
	Collard greens	129	32.2	[42]
	Kale	62	15.5	[43]
	Swiss chard	14	3.5	[44]
**Legumes**	Lentils	181	45.2	[45]
	Chickpeas	172	43.0	[46]
	Beans	149	37.2	[47]
	Peas	65	16.2	[48]
**Fruits**	Avocado	81	20.2	[49]
	Strawberries	50	12.5	[50]
	Papaya	37	9.2	[51]
	Oranges	30	7.5	[52]
**Whole grains**	Quinoa	42	10.5	[53]
	Oats	32	8.0	[54]
	Barley	19	4.8	[55]
	Brown rice	9	2.2	[56]

^1^ Considering 400 µg as the recommended daily intake of folate for teens and adults [57].

Fruits represent another category of natural sources of folate. Although fruits may not be as recognized as vegetables and legumes for their folate content, several varieties offer notable contributions to dietary folate intake [38]. Avocado, papaya, oranges, and strawberries present a satisfactory folate content, although in smaller amounts compared with other food groups [58]. Therefore, the incorporation of a variety of fruits into the diet provides natural sugars and flavor while contributing to folate intake.

In contrast with refined grains, which undergo processing steps that degrade some of their natural nutrients, whole grains, such as quinoa, brown rice, oats, and barley, retain their bran, germ, and endosperm, thus preserving essential vitamins and minerals, including folate. Therefore, such whole grains represent indispensable components for a balanced diet and important sources of folate. Their benefits go behind folate, also promoting adequate digestive and balanced health [59].

In addition to natural food sources, fortified foods are available with the addition of synthetic folic acid to prevent/restore folate deficiency. Such food sources include cereals, bread, pasta, and rice [37,60]. This strategy has been effective in reducing the prevalence of neural tube defects and improving folate status in populations worldwide [61,62].

Thus, the incorporation of these foods into the diet ensures optimal folate intake and reduces the risk of pathologic conditions related to folate deficiency.

Although both dietary folate and synthetic folic acid are bioavailable, they are absorbed and metabolized differently in the body, thus influencing folate homeostasis. One important aspect that influences the bioavailability of folate from dietary sources is the glutamate chain present in the molecule [63]. Folate in natural foods is mostly bound to a polyglutamate chain (minor proportion of folate in monoglutamate form; Figure 2). While the monoglutamate form is efficiently absorbed, polyglutamate folate is not. As such, the polyglutamate form of folate requires enzymatic decomposition to monoglutamate folate in the small intestine, by the enzyme glutamate carboxypeptidase II (GCPII; Figure 2), making it more efficiently absorbed by the intestinal cells. In contrast, synthetic folic acid present in fortified foods and supplements is in its monoglutamate form, favoring the intestinal absorption [63].

Another important aspect that influences the bioavailability of folate from dietary sources is the food matrix in which folate is present and food processing techniques. The food matrix can be defined as the cell structure that surrounds a nutrient and forms complexes with proteins and fibers. Studies comparing the consumption effects of whole spinach versus chopped spinach have shown an increased folate bioavailability, as traduced by increased plasma levels, upon disruption of the spinach matrix [39,64]. Additionally, food processing techniques can affect folate stability and bioavailability. A recent study investigated the influence of five different cooking methods—sous-vide cooking, microwave, cooking in a combi oven, steaming cooking, and boiling—on folate content in broccoli and spinach [65]. In spinach, all five methods resulted in a loss of folate content higher than 40% (sous-vide cooking: 41%; microwave 51%; cooking in a combi oven: 45%; steaming cooking: 57%; boiling: 61%). In contrast, folate content after cooking was better preserved in broccoli (loss of folate—sous-vide cooking: 23%; microwave 17%; cooking in a combi oven: 14%; steaming cooking: 28%; boiling: 52%) [65]. This indicates that different dietary sources of folate and cooking techniques influence folate bioavailability.

Other variables that actively influence folate bioavailability include genetic factors and putative nutrient interactions. MTHFR (the involvement of MTHFR enzyme in folate metabolism is detailed in Section 2, Folate-Dependent One-Carbon Metabolism: An Overview) and GCPII enzymes, which actively participate in folate metabolism (GCPII is responsible for cleaving polyglutamates from dietary folate [66]), show genetic variability [66,67,68]. The genetic polymorphism *C677T* in the *MTHFR* gene results in a substitution of an alanine by a valine and is associated with a decrease in enzyme activity by 30% in heterozygous individuals (CT) and 60% in homozygotes (TT), as compared with CC homozygous individuals [67]. The *H475Y* polymorphism in the GCPII-coding gene is associated with decreased plasma levels of folate [66,68]. Therefore, these genetic polymorphisms directly affect folate bioavailability.

On the other hand, the interaction with other vitamins/pharmaceuticals may greatly affect the folate cycle. Vitamin B12 is an indirect regulator of folate metabolism, as it influences the conversion of 5-MTHF to THF, the active form of folic acid [31,69]. A reduction in blood folate concentrations has also been associated with the use of oral contraceptives [70]. In a meta-analysis that included 2831 women (1359 users of oral contraceptives and 1472 controls) from 17 independent studies, plasma levels of folate were 1.27 µg/L lower in oral contraceptive users compared with control women. In addition, the combined analysis of folate concentration in red blood cells in 1389 women (772 users of oral contraceptives and 617 controls) from 12 independent studies revealed a reduction of 58.03 µg/L in oral contraceptive users compared to control women [70]. Therefore, this indicates that oral contraceptive users may be at higher risk of folate deficiency.

Treatment with the most common antiepileptic drugs has been linked to reduced plasma levels of folate during pregnancy [71,72,73]. In a total of 133 pregnancies from a group of 125 women with epilepsy, serum folate concentration showed an inverse relationship with phenytoin or phenobarbitone levels (from the eighth to the sixteenth week of pregnancy). However, no relationship was reported for carbamazepine [71]. The initiation of phenytoin treatment results in decreased serum levels of folate [72]. A reduction in serum folate levels and an increased incidence of folate levels below the normal physiologic range (4–15 ng/mL) have also been reported in patients treated with carbamazepine, gabapentin, oxcarbazepine, phenytoin, primidone, or valproate [73], and folate deficiency tends to be more pronounced in pregnant women using multiple anticonvulsants [74]. Nevertheless, lower levels of vitamin B12 were detected in patients under phenobarbital, pregabalin, primidone, or topiramate treatment, whereas treatment with valproate resulted in increased vitamin B12 serum levels [73]. This might suggest that anticonvulsants may indirectly perturb folate metabolism by interfering with vitamin B12 turnover. Overall, these data provide evidence that anticonvulsants can interfere with the folate cycle, potentially leading to suboptimal folate status.

The antiretroviral drug zidovudine alters thymidine metabolism, likely by inhibiting dihydropyrimidine dehydrogenase (DPYD-1), which results in dTMP depletion and folic acid accumulation [75]. Methotrexate, an antifolate drug used in cancer therapy, is a known inhibitor of DHFR, the enzyme that catalyzes the conversion of DHF into THF, which is required for the de novo synthesis of nucleotides for both DNA and RNA. In addition, methotrexate-polyglutamate also inhibits DNA synthesis by inhibiting TYMS, an enzyme required for the de novo synthesis of purines. These mechanisms underlie the use of methotrexate in cancer therapy [76]. The aminosalicylate sulfasalazine is a competitive inhibitor of intestinal folate transport, thus impairing folate absorption and metabolism [77,78]. However, in inflammatory bowel disease, it rarely results in folate deficiency, whereas in rheumatoid arthritis patients, the concentrations of folate in serum and red blood cells are frequently reduced [79]. The antibiotic trimethoprim, acting as an inhibitor of the DHFR enzyme, is also associated with a decrease in serum folate levels [80].

In summary, achieving optimal folate status may require a multifaceted approach, including dietary diversification, consumption of folate-rich foods, and supplementation when necessary. For individuals with specific genetic polymorphisms or with likely interactions with the nutrients or medicines that may impair folate metabolism, personalized interventions or targeted supplementation strategies may be required to optimize folate bioavailability.

## 4. Molecular Regulation of Folate Metabolism

The regulation of folate metabolism involves a myriad of factors that ensure the folate cycle is efficient and balanced to the cells’ needs. These factors include transcriptional, translational, and post-translational regulation of key enzymes, genetic polymorphisms, epigenetic modifications, availability of cofactors, and influences from other signaling pathways and cell cycle stages. The molecular mechanisms involved in the regulation of folate metabolism are illustrated in Figure 3.

### 4.1. Transcriptional, Translational, and Post-Transcriptional Regulation of Key Enzymes

As described above (see Section 2, Folate-Dependent One-Carbon Metabolism: An Overview), several key enzymes are involved in folate metabolism. The expression of genes encoding these key enzymes is regulated by transcription factors and signaling pathways, which are primarily influenced by the availability of folate and its derivatives and by the cell cycle stage.

Folate and its derivatives activate feedback mechanisms that modulate the expression of enzymes responsible for their metabolism. For example, the nutrient-sensing mechanistic target of the rapamycin (mTOR) pathway regulates cell growth in response to nutrient availability, including folate. When folate levels are abundant or low, mTOR can influence the activity of transcription factors, such as activating transcription factor 4 (ATF4) [81,82,83]. The transcription factors, in turn, promote the expression of proteins involved in nucleotide biosynthesis, including enzymes of folate metabolism [82]. These enzymes are then available to participate in folate metabolism to synthesize purines and pyrimidines, essential for DNA replication and cell proliferation.

Other transcription factors, such as E2F Transcription Factor 1 (E2F1) and Specificity Protein 1 (Sp1), hormones, including insulin, cell cycle proteins, like cyclin-dependent kinases (CDKs), nutrients, such as glucose and vitamin B12, and stress-related genes, such as Hypoxia-Inducible Factor 1 (HIF-1), can also trigger a positive or repressive response of genes that encode folate metabolism enzymes [84,85,86,87,88,89]. This transcriptional regulation occurs in coordination with the cell cycle, influencing both enzyme expression and localization. Different cell stages have varying demands for folate substrates, depending on whether the cells are proliferating or not. The mechanisms governing the expression and localization of folate cycle enzymes throughout the cell cycle, as well as the transcription factors involved, have been reviewed in the existing literature [90].

Translational regulation is another mechanism that governs the activity of folate cycle enzymes. Both DHFR and TYMS can autoregulate their synthesis by binding to their own mRNA, repressing translation [91,92]. In the case of DHFR, this autoregulation is relieved by the binding of its synthetic substrate, methotrexate (an antifolate drug), likely preventing unnecessary enzyme level increases when substrates or cofactors are unavailable. It has also been suggested that the translation of MTR mRNA is modulated by its cofactor, vitamin B12. However, to the extent of our knowledge, these initial findings have not been extensively investigated [93,94].

The enzymes of the folate cycle can also undergo post-translational regulations, such as phosphorylation and allosteric regulation. For instance, the MTHFR enzyme can be phosphorylated by kinases, such as casein kinases 1 and 2 (CK1/2), and its phosphorylation generally results in reduced activity [95,96,97,98]. This is because phosphorylation can cause conformational changes in the enzyme that decrease its ability to catalyze the reduction of 5,10-MTHF to 5-MTHF [96,98]. CK1/2 are ubiquitous kinases, common to several signaling pathways, that respond to cell cycle, cellular stress, and nutrient availability cues. Moreover, in vitro data suggest that MTHFR activity can also be modulated by the kinases DYRK1A/2 and GSK3A/B through phosphorylation [98]. Although DYRK1A/2 regulation is less understood and seems to be brain-related, GSK3A/B are inhibited by phosphoinositide 3-kinases (PI3K) upon insulin or other growth factor cues, highlighting the intricate interplay among various signaling pathways in the regulation MTHFR function [98,99]. Interestingly, PI3Ks are known upstream effectors of the mTOR (PI3K-AKT-mTOR pathway) that regulate cell proliferation [100]. Thus, it is possible that PI3Ks have the following dual action on folate cycle regulation: on the one hand, by inhibiting GSK3A/B, reducing MTHFR phosphorylation, and, consequently, increasing its activity, and, on the other hand, by activating mTOR signaling for the transcription factors that regulate the gene expression of folate cycle enzymes.

Besides phosphorylation, MTHFR is also allosterically inhibited by SAM, a product of the methionine pathway in 1C metabolism, and disinhibited by SAH, the demethylated form of SAM generated upon SAM’s reactions in the methionine pathway (see Section 2, Folate-Dependent One-Carbon Metabolism: An Overview) [98,101,102]. SAM and SAH compete for binding to MTHFR, so the SAM:SAH ratio is an important determinant of MTHFR activity within cells. Notably, phosphorylated MTHFR has been observed to be more sensitive to inhibition by SAM in in vitro studies, introducing an additional layer of complexity to its regulation [95,96,98].

Another example of allosteric regulation involves the interaction between MTHFS and the THF derivative, 10-formyltetrahydrofolate. Here, 10-formyltetrahydrofolate is able to bind to and allosterically inhibit MTHFS activity, serving as a feedback regulatory mechanism in the folate cycle [103].

Finally, the degradation of enzymes in the folate cycle can also function as a regulatory mechanism to maintain adequate levels of these enzymes in cells. The degradation pathways for these enzymes can vary. For instance, SHMT1 is known to be degraded by the ubiquitin–proteasome system, likely in response to cell cycle cues, while TYMS undergoes proteasome-mediated degradation in a ubiquitin-independent manner in the absence of ligand binding [104,105].

Overall, the expression, translation, and post-translational modifications of enzymes in the folate cycle must be tightly regulated to prevent depletion or excess of substrates crucial for essential cellular processes, such as DNA biosynthesis. This regulation is accomplished by a complex interplay among transcription factors, hormonal and nutrient signaling pathways, energy status, cell cycle, oxidative stress, hypoxia, availability of vitamin cofactors, and influences from phosphorylation, allosteric modulation, and degradation mechanisms.

### 4.2. Influence of Genetic Polymorphisms and Epigenetic Modifications

The expression of genes involved in the folate cycle is significantly influenced by common DNA variations, known as genetic polymorphisms, and by epigenetic modifications, such as DNA methylation. These collectively impact folate metabolism and, consequently, cellular function and disease susceptibility.

Among the most prevalent polymorphisms within the folate cycle are single nucleotide polymorphisms (SNPs) in the MTHFR gene, the most frequent and most studied being the C-to-T substitution at nucleotide position 677 (C677T) [67,106,107]. This substitution results in an alanine-to-valine change, leading to a significant reduction in the enzyme activity observed in vitro in both heterozygotes and homozygotes [67,108]. Its reduced activity leads to decreased folate and elevated HCY levels in the organism, a condition known as hyperhomocysteinemia, which is associated with an increased risk of cardiovascular diseases, congenital anomalies (such as neural tube defects), neurodegenerative disorders, and certain types of cancer [109,110,111,112]. Folic acid supplementation seems to counteract the MTHFR *C677T* effects on HCY and folate levels [113,114]. The MTHFR A-to-C substitution at position 1298 (A1298C) is another prevalent polymorphism, but most studies have shown mild or no implications of this variant on folate or HCY levels [114,115,116].

Several other genetic variants in folate cycle genes are described in the literature as having effects on enzyme activity, thereby impacting folate status and the metabolism of its derivatives. These polymorphisms are commonly associated with altered levels of folate, HCY, and/or other molecules involved in folate cycle metabolism, depending on the specific enzyme affected [111,113,117]. Consequently, they can lead to health risks similar to those associated with folate deficiency or hyperhomocysteinemia, as observed with MTHFR polymorphisms [110,111,118,119,120].

Along with genetic polymorphisms, epigenetic modifications add another layer of variability to folate cycle metabolism. Epigenetic modifications refer to changes in the genome that do not involve coding sequence modifications but can affect gene expression. These modifications include DNA methylation, histone modifications, and the activity of non-coding RNA molecules (e.g., microRNAs) [121,122,123]. They are influenced by various environmental and lifestyle factors, as well as transgenerational epigenetic inheritance, and can present unique patterns for each person [124,125]. Epigenetic modifications are crucial for normal development and cellular differentiation, and they play a role in the regulation of various biological processes (for example, DNA is methylated for X chromosome inactivation) [121]. Abnormal epigenetic modifications can lead to diseases, including cancer, neurological disorders, and cardiovascular diseases (reviewed in [9,126,127,128]). Unlike genetic mutations, epigenetic modifications are often reversible, making them potential targets for therapeutic interventions [129,130,131].

Folate cycle enzymes are intrinsically involved in epigenetic modifications, both regulating and being regulated by processes such as DNA and histone methylations. DNA methylation involves the addition of a methyl group (CH_3_) to the DNA molecule, usually resulting in the suppression of gene expression when it occurs in a gene promoter region. Folate is critical as a methyl group donor in the 1C metabolism pathway, and adequate folate levels are necessary for proper DNA methylation. Aberrant DNA methylation patterns, influenced by folate availability, can affect gene expression, including those involved in folate metabolism, thereby altering enzyme activity and various cellular processes. Consequently, abnormal methylation patterns due to folate status are linked to negative health outcomes, similar to those associated with genetic polymorphisms in folate enzymes, such as neurodegenerative disorders and neurodevelopment defects [132,133,134,135]. The interconnection between genetic polymorphisms and DNA methylation pathways within the folate cycle can lead to adverse health outcomes. For instance, the MTHFR *C677T* polymorphism, which leads to reduced blood folate levels and elevated HCY levels, has been correlated with lower methylation patterns [136]. This is not unexpected, given that MTHFR is the first enzyme in the DNA methylation pathway, playing a vital role in producing the necessary methyl groups for downstream molecules, such as SAM, to donate to DNA. While SAM is the direct methyl donor, its availability depends on the rate of conversion of HCY to methionine, which in turn depends on upstream methyl donors that are produced by MTHFR. Therefore, abnormal function of MTHFR leads to lower levels of folate derivatives and decreased DNA methylation. In line with this, studies indicate that folate intake and availability in mammals impact the methylation and demethylation patterns of the genome during genetic imprinting. This may have a significant impact on later-life health, suggesting a role of maternal diet in normal DNA methylation patterns during fetal development [137,138,139,140]. However, studies on the effects of folic acid supplementation on DNA methylation in adults show inconsistent results regarding whether methylation increases, decreases, or remains unchanged. These variations may be attributed to differences in supplementation periods, specific genes analyzed, and biological sample types, as different tissues exhibit distinct methylation patterns and requirements [141,142,143,144]. Overall, is it well established that the folate cycle impacts DNA methylation by providing methyl groups to DNA. This process is particularly important during rapid cell proliferation stages to maintain methylation levels in every new cell. However, the underlying regulatory mechanisms controlling DNA methylation and consequent expression patterns in each tissue remain incompletely understood. The expression of folate cycle genes is also regulated by DNA methylation. For instance, folate receptor 1 (FOLR1), a key protein involved in folate uptake, is subject to epigenetic regulation. It was found that the placenta of preterm newborns had lower levels of the FOLR1 gene, which was inversely correlated with increased methylation of the gene [145]. Also, the MTHFR gene was found to exhibit different methylation patterns according to lifestyle factors such as physical activity, disease context like diabetes or pre-eclampsia, and age [146,147,148,149]. This dual influence of DNA methylation—both regulating and being regulated by folate cycle enzymes—highlights the intricate interplay between genetics and epigenetics in maintaining cellular function and overall health.

Another type of epigenetic alteration associated with the folate cycle is histone modification, including methylation, acetylation, phosphorylation, and ubiquitination. Histones are proteins that package and stabilize DNA, forming a structure called chromatin. Therefore, histone modifications can influence the chromatin structure and accessibility of DNA to the transcriptional machinery, impacting gene expression, including that of folate metabolism-related genes [150].

In the case of histones, methylation can either repress or activate gene expression, depending on the methylated histone residue and the extension of the methylation. Like DNA methylation, histone methylation status may be influenced by the availability of methyl donors from the folate cycle supply. However, folate appears to play roles beyond carrying carbon units. It has been found to act as an enzymatic cofactor for the histone lysine demethylase (LSD1), an enzyme involved in histone demethylation. Low folate levels have been linked to increased histone methylation in certain genes, likely because of reduced LSD1 activity in a folate-deficient environment [151,152].

Although studies are limited, there is also evidence that histone modifications play a role in regulating the expression of folate cycle genes, alongside genetic, epigenetic, transcriptional, and post-translational factors. For example, the expression of the DHFR gene is regulated by chromatin remodeling, involving the acetylation and methylation of histones at its promoter region. These modifications are known to occur in a cell cycle-dependent manner, likely because of the varying needs for DNA synthesis at different stages of the cell cycle [153,154].

In summary, there is a complex interplay between folate cycle activity and genetic and epigenetic factors, where folate metabolism both regulates and is regulated by these factors. Understanding this interplay is crucial for assessing disease risk and developing targeted interventions.

## 5. Folate Metabolism and Cancer

### 5.1. The Importance of Folate Metabolism in Cell Division and Its Dysfunctional Regulation in Cancer

The importance of folic acid in cell division has been well-documented for more than half a century [155,156]. Folate metabolism plays a critical role in the synthesis and repair of DNA by providing the necessary 1C units for the biosynthesis of purines and pyrimidines, the essential bases that compose nucleic acids. For purines, 1C units from folate derivatives, in the form of formyl groups, are used by specific enzymes to form the purine ring. For pyrimidines, THF donates a methylene group for enzymes that convert dUMP, which contains the RNA base uracil (U), to thymidylate (dTMP), which contains the DNA base thymine (T), through methylation [157]. Thus, adequate folate levels ensure proper DNA synthesis and replication, maintaining genomic stability, which is essential for a successful cell division (Figure 4). During cell division, a sufficient supply of folate is vital to prevent DNA mutations and chromosomal abnormalities, which can lead to cancer development [158,159]. Folate deficiency results in the extensive incorporation of uracil bases into human DNA, likely because of deficient methylation of uracil to thymine. Attempts to repair the misincorporation of uracil in DNA lead to an increased rate of chromosome breaks. Both the elevated levels of uracil and the chromosome breaks in DNA were reversed by folate administration [158]. Insufficient folate levels can also compromise the proper methylation of certain genes, such as oncogenes, whose expression or overexpression can induce carcinogenesis (Figure 4) [159]. Although cancer cells may face limited proliferation rates on low folate levels, because of a lack of supply for DNA replication, folate deficiencies and the consequent reduction in purine and pyrimidine biosynthesis particularly affect rapidly dividing cell types. This was observed, for instance, in hematopoietic cells, leading to reduced blood cell proliferation and the development of conditions like anemia, leukopenia, or thrombocytopenia [160,161]. The World Health Organization (WHO) states that the risk of developing anemia is high when serum folate concentrations are below 3 ng/mL and red blood cell folate levels are under 100 ng/mL [162,163]. This underscores the importance of consuming adequate folate through a varied diet or appropriate supplementation.

Despite some clear evidence, the relationship between folate and cancer risk is not translatable to all types of tumors. For instance, while folate intake has a protective effect against certain cancers, in some tumors, it has minimal to no impact, and in other cases, higher folate consumption may increase the risk of cancer development [22,164,165,166,167,168].

Under certain circumstances, exceeding the recommended daily intake of folates could potentially be deleterious to human health. Dietary supplementation with folic acid has been shown to accelerate the growth of acute lymphoblastic leukemia, suggesting possible adverse effects such as cancer progression [169]. Farber’s study observed that supplementation with folic acid could counteract the effects of methotrexate in leukemia cells by saturating the enzymes targeted by methotrexate, reducing its effectiveness [169]. Other studies suggest that high folate levels may exacerbate the carcinogenic effects of smoking, especially in heavy smokers [164]. Research conducted in northern Poland involving 132 lung cancer patients found that elevated serum folate concentrations above the median (>17.5 nmol/l among healthy controls) were associated with a higher risk of developing lung cancer. Specifically, subjects with higher serum folate levels had a higher tendency for lung cancer [164]. Folate protects normal cells from mutations, but once initial DNA damage occurs, elevated folate levels can tip the balance towards the progression of neoplastic cells into cancer by promoting the increased synthesis of nucleic acids [164,170]. Another study published in 2007 described an association between folic acid supplementation in the United States and an increase in colorectal cancer rates [165]. Pooled findings from two randomized, placebo-controlled clinical trials indicated that supplementation with folic acid and vitamin B12 was linked to increased rates of colorectal cancer incidence because of aberrant DNA methylation [171,172]. 

The meta-analysis conducted by Bo et al. revealed that folate is associated with an increased risk of prostate cancer [173]. While prostate cancer cells typically divide more slowly compared with other types of tumors, the prostate depends significantly on the folate 1C metabolism pathway to produce polyamines from s-AdoMet. Normally, the growth of these cells is constrained by low folate levels. However, exposure to higher concentrations of folate, such as through supplementation, can provide transformed cells with a potential advantage in proliferation [174]. When prostate cancer cells were compared in environments with varying levels of folic acid, those exposed to lower concentrations exhibited notable genetic and epigenetic instability along with observable phenotypic changes. Specifically, between 24% and 37% of cells showed chromosomal rearrangements, and there was increased CpG island hypermethylation compared with cells in environments with higher folic acid concentrations [175].

Conversely, a different study conducted in Finland found no correlation between serum folate levels and changes in lung cancer risk [166]. In a study conducted by Pitsick and colleagues that used a mouse model of oral cancer, dietary folate had no effect on total tumor burden, histopathology, folate transporter expression, or lymphoid cell abundance [176].

On the other hand, folate intake is associated with a protective effect for some cancers. Low plasma folate levels may increase the risk of hepatocellular carcinoma by inducing DNA hypomethylation, which can lead to the dysregulation of proto-oncogenes and tumor suppressor genes [177,178]. Folate deficiency or impaired folate metabolism is associated with elevated HCY levels, DNA hypomethylation, DNA damage, disrupted cell proliferation, malignancies, and reduced nitric oxide production [179,180]. Falling below a certain threshold of folate status can result in mitochondrial dysfunction and a ROS-induced vicious circle, which activate apoptotic signaling and cellular death, collectively contributing to a carcinogenic mechanism [181]. The correlation between a low intake of vegetables and fruits and an increased risk of head and neck squamous cell carcinoma (HNSCC) implies that folate, found abundantly in these food groups, acts as a protective nutrient against HNSCC [182]. Indeed, some meta-analyses indicate a significant association between increased folate intake and a nearly 50% decreased risk of head and neck squamous cell carcinoma [167], as well as esophageal and pancreatic cancer [168] and breast cancer [22]. In these types of cells, folate deficiency reduces de novo thymidylate biosynthesis, leading to uracil mis-incorporation during DNA repair and synthesis, which causes DNA strand breaks and ultimately leads to malignant transformation [158,167,183]. Thus, it is important to note that despite the type of cancer, the relationship between folate intake and cancer risk can also vary depending on individual factors such as genetic predisposition, overall diet, and lifestyle habits.

Nilsson et al. showed that both normal and cancer cells express THF, MTHFD1, and MTHFD2, although MTHFD2 generally displays a higher baseline expression [184]. MTR activity can influence the overall levels of methionine and SAM in the body. Multiple pieces of evidence indicate that the involvement of the enzyme MTR in the folate cycle could be crucial for tumor development by specifically disrupting the production of these vital metabolites [185]. When MTR activity is compromised by defective vitamin B12 binding or availability, it disrupts the folate cycle and hinders the rapid proliferation of cancer cells [185]. Likewise, MTR exhibits widespread expression across human cancer cell lines and tumors [186,187]. Nevertheless, the broader metabolic impacts of disrupting MTR activity have not been explored within normal folate conditions. Thus, it remains unclear whether deliberately reducing MTR expression would inhibit tumor growth in animals [185,188].

Additionally, several studies have identified ALDH1L1 as a potential tumor suppressor [189,190,191]. ALDH1L1 may inhibit tumor growth by reducing the availability of 10-formyl-THF, a critical component for purine synthesis, thereby limiting DNA synthesis and cell proliferation. Thus, as an adaptive response, most cancer cells show a downregulation or loss of ALDH1L1 [192,193], which results in an increased pool of 10-formyl-THF, supporting enhanced nucleotide biosynthesis and tumor progression.

### 5.2. Pharmacological Agents Targeting Folate Metabolism in Cancer

The concept that folate antagonists could restrict the growth of malignant cells boosted the development of modern cancer treatment, particularly the category of medications called antifolates, which include methotrexate, pralatrexate, pemetrexed, and raltitrexed. Since then, methotrexate, arguably the most recognized antifolate, has been employed in the treatment of various neoplastic and inflammatory conditions [28]. Mechanistically, methotrexate inhibits several enzymes that participate in nucleotide synthesis, including DHFR, TYMS, aminoimidazole caboxamide ribonucleotide transformylase (AICART), and amido phosphoribosylltransferase. The restricted use of methotrexate is associated with resistance mechanisms that cancer cells develop to reduce the efficacy of antifolates [194]. Pralatrexate is another antifolate drug that inhibits the enzyme DHFR and is primarily used in relapsed or refractory peripheral T-cell lymphoma [195,196]. Pemetrexed inhibits TYMS, THF reductase, and glycinamide ribonucleotide formyltransferase and has proven efficacy for non-squamous non-small cell lung cancer [197]. Raltitrexed works by inhibiting TYMS and is primarily used in advanced colorectal cancer [198]. In addition, 5-fluorouracil (5-FU) specifically targets TYMS and is commonly used as a primary chemotherapy option for colorectal cancer, typically resulting in response rates of around 60–65% [199,200]. Inhibitors targeting purine or pyrimidine synthesis have also been utilized in hematological malignancies. For instance, 6-mercaptopurine (6-MP), a thio-substituted purine analog that inhibits de novo purine synthesis, is frequently used in combination with methotrexate as maintenance therapy in childhood acute lymphoblastic leukemia [201].

Table 2 summarizes the mechanisms of action as well as the clinical applications of the various antifolate drugs available in cancer therapy.

MTR expression is indispensable for cancer cells to initiate tumor growth in mice and can influence their responsiveness to antifolate medications. Inhibiting MTR could potentially serve as a viable alternative for cancer treatment, prompting a reassessment of drug development efforts. However, successful targeting of MTR would necessitate that tumor cells exhibit a greater need for MTR activity compared with other tissues [185]. Thus, considering the significant effectiveness of antifolate therapies in cancer and the crucial role of MTR in sustaining tumor folate levels, exploring the targeting of MTR, either alone or in conjunction with other antifolates, is an area that needs future exploration to understand its potential effectiveness in cancer treatment.

FOLR1, a membrane glycoprotein responsible for transporting reduced folates and folic acid into cells [202], represents a promising target for anticancer therapies. In fact, FOLR1 is found in abnormal quantities in various epithelial tumors, including ovarian, lung, and breast cancer, enhancing the likelihood of ligand binding to the tumor [203,204,205]. Its safety as a target results from its restricted expression in non-cancerous cells as opposed to its high expression in tumor cells [206]. Thus, actively targeting folate receptors with a ligand is a promising approach to enhance the selectivity of therapeutic agents [205].

Targeted radionuclide therapy has become a promising option for the palliative treatment of metastatic cancer [207,208]. Folic acid radioconjugates have been adapted for clinical use to perform nuclear imaging on FOLR-positive tumors [209,210]. The issue of high renal uptake of folate-based radiopharmaceuticals has been tackled using various strategies, including pharmacological interactions [211,212]. In fact, the high renal uptake of folate-based radiopharmaceuticals is a significant issue as it can limit the treatment’s effectiveness and increase side effects in the kidney. A significant advancement was made by incorporating an albumin-binding component into the structure of radiofolates to extend their circulation time in the bloodstream [213,214]. This modification aims to prolong the circulation time of radiofolates in the bloodstream, allowing them more time to reach and be absorbed by tumors, which is crucial for both tumor detection (imaging) and treatment (therapy) [213].

**Table 2 ijms-25-09339-t002:** Comprehensive overview of pharmacological agents targeting folate metabolism including mechanisms of action, clinical applications, and other observations.

Antifolate Drug	Mechanisms of Action	Clinical Applications	Observations	References
**Methotrexate**	Inhibits DHFR, TYMS, AICART, and amido phosphoribosylltransferase	Various types of cancer	Rapid development of resistance mechanisms by cancer cells	[194]
**Pralatrexate**	Inhibits DHFR	Relapsed or refractory peripheral T-cell lymphoma	Patients should receive folic acid vitamin B12 supplementation	[195,196]
**Pemetrexed**	Inhibits TYMS, THF reductase, and glycinamide ribonucleotide formyltransferase	Non-squamous non-small cell lung cancer	Effective treatment	[197]
**Raltitrexed**	Inhibits TYMS	Advanced colorectal cancer	Effective substitute for 5-FU in metastatic gastric cancer	[198,215]
**5-FU**	Inhibits TYMS	Colorectal cancer	Response rates of 60-65%	[199,200]
**6-MP**	Inhibits de novo purine synthesis	Various types of cancer	Used in combination with methotrexate	[201]

**AICART**, aminoimidazole caboxamide ribonucleotide transformylase; **DHFR**, dihydrofolate reductase; **THF**, tetrahydrofolate; **TYMS**, thymidylate synthase; **5-FU**, 5-fluorouracil; **6-MP**, 6-mercaptopurine.

### 5.3. Preclinical and Clinical Studies Evaluating the Efficacy and Safety of Folate-Targeted Therapies in Cancer

Tailored nutritional strategies hold significant potential in the fight against cancer by targeting the unique metabolic requirements of tumor cells. For instance, in a proof-of-principle clinical trial, six healthy middle-aged individuals were enlisted and placed on a low methionine diet, with an intake of approximately 2.92 mg/kg/day, equivalent to an 83% reduction in daily methionine consumption, for three weeks. By interrupting the flow of 1C metabolism through the methionine-reduced diet, cell vulnerabilities emerged, particularly in redox status and nucleotide metabolism, similar to those observed in mouse models. In mice, these vulnerabilities resulting from low methionine, combined with other therapies, such as radiation and antimetabolite chemotherapy, that also specifically target these aspects of cancer metabolism, led to reduced tumor growth. Because of the conservation of these metabolic processes, similar outcomes are expected in humans, providing a new strategy to improve anticancer therapeutic efficacy [216].

Along with dietary interventions, numerous therapeutic agents targeting FOLR are actively being evaluated in clinical trials. These treatments take advantage of the overexpression of FOLR in certain cancers to deliver targeted therapies. These approaches focused on FOLR1 have shown effectiveness in clinical settings with minimal adverse effects [217].

FOLR-targeted small-molecule conjugates have also shown promising results in early-stage clinical trials with microtubule destabilizing agents. Vintafolide (EC145), a conjugate of folic acid with the vinca alkaloid desacetylvinblastine hydrazide, initially showed promise when used alongside docetaxel in non-small cell lung cancer patients [218]. However, a phase III trial investigating EC145 in combination with doxorubicin hydrochloride liposome for platinum-resistant ovarian cancer patients was suspended, possibly because of its inability to improve progression-free survival [219]. Pafolacianine (OTL38), another FOLR-targeting therapeutic, is currently in development and undergoing clinical trials [220].

Reddy et al. developed EC2629, a folate conjugate of a DNA crosslinking agent that features a novel DNA-alkylating component. This compound demonstrated remarkable potency, and treatment with EC2629 in nude mice with FOLR-positive KB human xenografts resulted in 100% cure rates at very low doses [204].

The initial monoclonal antibody therapy directed at FOLR1, known as Farletuzumab (MORAb003), was developed and successfully administered to patients without safety concerns. However, this approach did not demonstrate effectiveness in reducing tumors [221]. Numerous antibody-drug conjugates (ADCs) employing this strategy are currently in clinical trials [205]. MORAb202 is an ADC consisting of a FOLR-binding antibody, and it was well-tolerated in patients with advanced solid tumors in a phase 1 study [222]. Another ADC targeting FOLR1 for ovarian cancer is IMGN853, also known as Mirvetuximab soravtansine (IMGB853) or Elahere™ [223,224]. However, this approach requires high expression levels of the target antigen, targets only dividing cells with the cytotoxic payload, and its anti-tumor effectiveness may not last after treatment stops [225].

Alternatively, cellular immunotherapy presents a promising targeted approach. Genetically modifying T cells to express chimeric antigen receptors (CARs) has demonstrated efficacy. Initial studies involving genetically modified T cells targeting FOLR1 were safe for patients but did not show anti-tumor effectiveness [226,227]. Daigre et al. reported promising preclinical data on a novel FOLR1-directed CAR T cell candidate that induces rapid tumor eradication, CAR T cell proliferation, and short-term persistence [225].

Regarding a new class of albumin-binding radioconjugates, Guzik et al. evaluated this system using 5-MTHF as the targeting agent. In a preclinical study, they demonstrated the promising potential of 5-MTHF-based radioconjugates for targeting FOLR. Their study showed slower tumor growth and, consequently, an increased median survival time [228]. Table 3 summarizes the description as well as the clinical applications of the different antifolate drugs.

Concerning folate-functionalized nanoparticle drug delivery systems, it appears that none have successfully completed any clinical trial stages thus far [205].

In summary, present strategies for targeting folate metabolism in cancer predominantly center on dietary manipulation and the targeting of folate receptors.

### 5.4. Impact of Dietary Modifications on Cancer Risk

Nutrition and lifestyle significantly influence cancer development, with estimates suggesting that inadequate dietary habits may contribute to over one-third of cancer-related fatalities. The nutrient composition of growth media significantly impacts cancer cell metabolism [237,238,239]. However, the degree to which diet, by affecting circulating metabolite levels, mimicking the in vivo scenario, alters metabolic pathways in tumors and influences therapeutic outcomes remains largely unexplored. Research indicates that dietary adjustments, such as removing amino acids like serine and glycine, can influence cancer prognosis [240,241]. As referred, a promising approach for a focused dietary intervention in cancer involves limiting methionine, an indispensable amino acid crucial in 1C metabolism [216].

While MTR’s contribution to overall methionine production seems minimal, several lines of evidence highlight its essential role in replacing the THF backbone from 5-methyl THF, crucial for nucleotide synthesis. Dietary restrictions on methionine, known to impede tumor growth, can synergize therapeutically with inhibiting nucleotide synthesis in certain cancer types [185,216,242]. Research has shown that restricting dietary methionine can inhibit the de novo synthesis of methionine, likely indicating decreased MTR activity [216]. The findings described by Sullivan et al. propose a potential mechanism by which methionine limitation might synergistically affect tumor growth alongside decreased nucleotide synthesis. Reduced MTR activity could lead to limited THF regeneration, further impeding nucleotide synthesis [185]. However, although animal models have shown promising anticancer effects with methionine restriction, clinical studies employing methionine-restricted diets remain limited. A more hopeful approach could involve the utilization of a methionine-depleting enzyme, such as methioninase [242].

## 6. Folate Metabolism and Neurodegeneration

### 6.1. Roles of Folate Metabolism in Neural Tube Formation, Neuronal Function, and Neurotransmitter Synthesis

Folate metabolism is critical in the context of neurodevelopment and neural function. Early reports linked low folic acid levels in pregnant women to a higher incidence of pregnancy complications, including miscarriage, and fetal malformations [243]. This subject was thoroughly investigated over the years until it became widely recognized that during early embryogenesis, folate is essential for the formation of the neural tube, the precursor to the central nervous system (CNS) [244,245]. Folate deficiency, whether from a nutritional deficit or genetic polymorphisms, during this critical period of rapid cell division, can result in severe neural tube defects (NTDs) such as spina bifida and anencephaly. NTDs result from the incomplete closure of the neural tube during its formation in early embryogenesis (21–28 days post-conception). They are among the most common congenital malformations, representing a significant public health issue associated with miscarriages, mortality (inevitable in cases of anencephaly), morbidity, and high socioeconomic costs [245,246].

The underlying mechanism linking folate deficiency to NTDs has been the subject of extensive research for many years, yet it remains incompletely understood. The most widely accepted theory is the “methylation hypothesis”. This hypothesis is supported by data on various biological factors that can result in NTDs, with methylation emerging as a common underlying mechanism. One of the first links between methylation and NTDs comes from the strong correlation between elevated HCY levels and increased risk of NTD [247,248]. This connection highlights the central role of genetic factors and the consequent impairment of the methylation pathway in the development of NTDs. The identification of the MTHFR *C677T* polymorphism as a genetic risk factor for NTDs bolstered this hypothesis [249]. This polymorphism is known to cause elevated HCY levels due to the deficiency in the production of the methyl donor 5-MTHF by MTHFR. The lack of 5-MTHF impairs the remethylation of HCY to methionine, which is subsequently converted into SAM, the primary methyl donor for DNA methylation. Other data have shown that perturbations in DNA methyltransferases, the enzymes responsible for DNA methylation, during embryonic development in mice result in NTDs [250]. Similar observations were made in chicken and rat embryos with disturbed DNA methylation. Interestingly, the administration of HCY to rats with low methionine levels prevented NTDs by restoring the methylation pathway. This highlights that elevated HCY levels are not the direct cause of NTDs but rather a biomarker for a disturbed methylation cycle and high NTD risk. This also explains why the administration of HCY to pregnant mice or mouse embryos did not induce NTDs and why the analysis of other polymorphisms in genes that regulate HCY accumulation did not associate with NTD risk in humans or mouse models (reviewed in [244,251]). These findings underscore the following key points: (1) the importance of methylation during rapid cell growth in early embryogenesis and (2) the hypothesis that the MTHFR *C677T* variant increases NTD risk through impaired methylation rather than through HCY toxicity or other mechanisms.

More recently, studies have linked folate deficiency to hypomethylation of genes in several molecular pathways, whose impairment can contribute to the onset of NTDs. For instance, low folate levels induced hypomethylation and consequent activation of histone ubiquitination genes. This activation led to the downregulation of genes involved in neural tube closure in mouse models [252]. Altered expression of folate transporters was also linked to reduced methylation and abnormal expression of the *SOX2* gene involved in embryonic development and cell fate in the dorsal neural tube of chicken embryos. This disruption impairs normal neural crest development and has potential implications for the development of NTDs [253]. Indeed, studies have found an association between folate receptor genetic variants or polymorphisms and embryonic malformations, including NTDs, although the underlying mechanism was elusive [254,255]. The involvement of cytoskeleton proteins has also been studied, revealing that low folate levels cause impaired methylation of septins, which are essential for proper neural tube closure [256]. Mitochondrial DNA de novo methylation is crucial in protecting the mitochondrial genome against oxidative stress. This protection is essential because mitochondria produce high levels of reactive oxygen species (ROS) during the high-demand metabolism of early embryonic development. Therefore, a lack of folate may impair this protective methylation process, increasing the risk of defective early embryogenesis and resulting in malformations such as NTDs [257].

Apart from DNA methylation, other genome defects have been studied. For instance, a cohort study found that NTD cases displayed modifications in histones that recruit DNA mismatch repair machinery to the genome. The same study observed that, in human neuroectodermal cells, folate deficiency was associated with the disrupted recruitment of histones and DNA repair machinery to the required neural tube closure genes, leading to the accumulation of rare genetic variants during neural tube closure, which contributed to the onset of NTDs [258]. Other molecular mechanisms and genetic polymorphisms linking genome and epigenetic changes in response to folate metabolism to the development of NTDs were recently reviewed [131,133]. Investigations into whether the association between low folate levels and impaired DNA biosynthesis, rather than impaired DNA methylation, was the underlying cause of malformations in early embryogenesis were also made but have not conclusively demonstrated a clear link to NTDs in humans [251].

There was a significant advance in the prevention of NTDs with the acknowledgment that these conditions are substantially reduced by maternal folic acid supplementation taken pre-conceptionally and during pregnancy [245,259]. This was observed in every country that implemented large-scale food fortification or supplementation with folic acid [245,260,261,262]. Therefore, global WHO guidelines recommend that the optimal red blood cell folate concentration to reduce NTDs effectively is above 400 ng/mL (906 nmol/L) in women of reproductive age [162]. To maintain those levels, current guidelines recommend the intake of 400 µg of folic acid per day or higher for people of fertile age or with altered levels of biomarkers related to 1C metabolism, such as elevated HCY (different guidelines can be consulted in Appendix A, Table 2 from [263]).

Overall, there is a strong correlation between folate deficiency and NTDs, with the underlying mechanisms likely involving a combination of several interconnected pathways.

Along with its critical role in neurodevelopment, folate metabolism also supports neuronal function and homeostasis throughout life. Folate participates in the biosynthesis of monoamine neurotransmitters, including serotonin, dopamine, and norepinephrine [264]. These neurotransmitters are essential for mood regulation, appetite control, sleep, cognitive function, and overall mental health [265]. SAM, the downstream metabolite of methionine, is the sole methyl donor in the CNS and is involved in the donation of methyl groups necessary for the formation of tetrahydrobiopterin (BH4), which is a cofactor in the reactions that convert amino acids into monoamine neurotransmitters. When folate metabolism is deficient, levels of SAM and neurotransmitters in the cerebrospinal fluid (CSF) are decreased, which is associated with neuropsychiatric conditions such as depression [266,267]. Although the pathogenesis of depression is complex and multifactorial, one of the most widely accepted underlying mechanisms for depression is the “monoamine hypothesis,” which posits that low levels of monoamine neurotransmitters are at the root of depressive symptoms (recently reviewed in [268]). It is known that individuals with depression often present low folate blood levels [269,270]. Low serum folate levels are associated with a lack of response to antidepressant drugs, and adding folate supplementation improves overall antidepressant response (reviewed in [271]). Reduced forms of folic acid, such as SAM and methylfolate, are the preferred forms utilized by the brain and have demonstrated some efficacy in improving depressive symptoms [272,273,274]. However, studies on this topic remain controversial, with varying results regarding folate effectiveness [275,276].

Folate metabolism dysregulation has also been linked to other neuronal pathologies, such as autism spectrum disorders and refractory schizophrenia [277,278]. A common feature among these conditions is cerebral folate deficiency due to the presence of folate receptor autoantibodies, which block the transport of folate to the CSF and brain. The intake of a reduced form of folate, folinic acid (also called leucovorin), which is readily converted into THF without requiring DHFR, has been shown to improve autistic features in children, such as communication and attention, and to stabilize schizophrenia symptoms [277,278,279]. This conversion bypasses the effects of drugs or genetic polymorphisms that inhibit DHFR. Individuals carrying the common MTHFR *C677T* polymorphism are associated with a higher risk of developing neuropsychiatric disorders, such as depression, schizophrenia, and bipolar disorder [280,281]. The presence of folate receptor autoantibodies has also been associated with brain inflammatory diseases, such as multiple sclerosis [282]. Brain inflammation is an expected consequence of folate deficiency or impaired metabolism since these lead to the accumulation of HCY, which is associated with neurotoxic events. These events include mitochondrial dysfunction, with consequent oxidative stress, loss of calcium homeostasis leading to brain cells’ death, and accumulation of hyperphosphorylated proteins, such as tau and β-amyloid, as observed in vitro and in animal models [283,284,285,286,287,288,289] (Figure 5). These pathophysiological changes suggest that impaired folate metabolism could be a contributing factor to the development of neurodegenerative diseases, such as Alzheimer’s disease.

In summary, proper folate metabolism is essential for neuronal homeostasis throughout life. Inadequate folate metabolism can disrupt essential neuronal processes, potentially contributing to neurodevelopmental defects, neuropsychiatric disorders, and, later in life, neurodegenerative diseases (Figure 4).

### 6.2. Folate Metabolism in Neurodegeneration

Neurodegenerative diseases are a growing concern worldwide, with conditions such as Alzheimer’s disease (AD) and Parkinson’s disease (PD) becoming more common [290]. Lifestyle, environmental, and genetic factors, along with aging, can all contribute to neurodegeneration [291]. Nutrition also seems to play a role, as both dietary patterns and specific nutrients, such as omega-3 fatty acids and B vitamins like folate, have shown the potential to protect against cognitive decline [292,293].

B vitamins are involved in the metabolism of HCY, and high serum levels (hyperhomocysteinemia) are recognized as a risk factor for cognitive impairment and dementia. HCY can be metabolized via the following distinct routes: the vitamin B12 and folate-dependent remethylation pathway, which regenerates methionine, and the vitamin B6-dependent transsulfuration pathway, which converts HCY into cysteine. The imbalance between these metabolic pathways results in an elevation of HCY levels in the bloodstream, leading to hyperhomocysteinemia [294]. In a 2022 meta-analysis, Wang et al. reported a correlation between B vitamin supplementation and delayed cognitive deterioration, especially when administered early and over a prolonged period. Moreover, inadequate folate levels, detected even in individuals with mild cognitive impairment (MCI), were recognized as factors predisposing to dementia and cognitive decline. Conversely, increased folate intake was associated with a reduced susceptibility to dementia in non-demented populations [295] (Figure 5).

### 6.3. The Role of Folic Acid Supplementation in Neurodegeneration

Research indicates that individuals with neurodegenerative diseases have lower levels of folate and vitamin B12, and higher levels of HCY, compared with control subjects [296,297,298,299,300,301,302] (Figure 5). Clinicians commonly prescribe folic acid supplements for patients with cognitive impairment and neurodegenerative diseases; however, their effectiveness remains controversial.

Some clinical trials point to a beneficial impact of folic acid supplementation on cognitive performance and HCY levels in patients with folate deficiency and cognitive impairment. In 2001, Yukawa et al. described that folate therapy (15 mg/day) improved neurological symptoms in 67% of folate-deficient patients (24 out of 36 cases) after 60 days [299]. Accordingly, Ma et al. reported that folic acid supplementation (400 μg/day) may benefit individuals with MCI by improving cognitive performance and possibly influencing disease-related biomarkers and inflammatory processes associated with cognitive decline (e.g., decrease in HCY and TNFα levels) [303,304,305]. Previously, Durga et al. reported that 3 years of oral supplementation with folic acid (800 μg/day) improved memory, information processing speed, and sensorimotor speed in adults aged 50–70 with high HCY levels, suggesting cognitive benefits [306]. Moreover, in senescence-accelerated mouse prone 8 (SAMP8) mice, folic acid supplementation (2.5 and 3.0 mg folic acid/kg diet) significantly delayed age-related neurodegeneration and cognitive decline by reducing reactive oxygen species and enhancing mitochondrial function through the telomere-p53–mitochondria pathway, suggesting its potential in mitigating brain aging effects [307].

Despite this promising evidence, other clinical trials have shown less enthusiastic results. Sommer and colleagues found no significant cognitive improvement in dementia patients under supplementation with 10 mg/day of folic acid. In fact, there was a trend suggesting a potential worsening of certain cognitive abilities in the group receiving folic acid [308]. Research into AD and PD has also investigated the potential of folic acid supplementation to alleviate cognitive decline and other symptoms associated with these neurodegenerative disorders [309,310,311,312,313].

Chen et al. followed patients aged 40–90 with mild to moderate AD, treated with donepezil (5 mg daily, increased to 10 mg after one month) under folic acid supplementation (1.25 mg/day) over 6 months. Folic acid supplementation increased the Mini-Mental State Examination (MMSE) score, suggesting improved cognition (Figure 5). Moreover, folate and SAM levels were significantly higher, while HCY, Aβ_40_, presenilin 1-mRNA (PS1, important role in Aβ generation), and TNFα-mRNA levels were lower in folate-supplemented individuals [309]. More recently, Hama et al. showed that folate supplementation (5 mg/day) significantly decreased HCY levels while improving average MMSE scores [311].

Connelly et al. also demonstrated a positive synergistic effect between folic acid supplementation (1 mg/day, for 6 months) and the effectiveness of cholinesterase inhibitors used in AD treatment [310] (Figure 5). In contrast, in a study involving 466,224 participants from the U.K. Biobank, Ling et al. associated folate/folic acid supplementation with an increased risk of AD and vascular dementia, as well as adverse changes in hippocampal and amygdala volume. However, the combination of folate/folic acid with other B vitamins mitigated the elevated risk of AD and vascular dementia associated with folate/folic acid supplementation alone [313].

Muller et al. reported significantly higher HCY levels in levodopa-treated PD patients compared with untreated patients and controls [314]. However, it is important to note that levodopa treatment may deplete methyl groups, raise HCY levels, and potentially accelerate neurodegeneration [315]. In a study involving 30 PD patients, Ibrahimagic et al. revealed hyperhomocysteinemia in 20% of the patients, which was effectively normalized with periodic oral folic acid administration (15 mg/day for 1–2 months) over five years. This suggests that folic acid could become a preferred treatment for hyperhomocysteinemia in patients with PD [312] (Figure 5). The results obtained in the clinical trial were further supported in animal studies [316,317,318]. Indeed, in the 1-methyl-4-phenyl-1,2,3,6-tetrahydropyridine (MPTP)-induced mouse model of PD, Jia et al. reported that supplementation with folic acid (5 mg/kg of body weight) significantly improved motor performance and reduced levels of HCY and oxidative damage in the substantia nigra [316]. Additionally, in a rat model of 6-hydroxydopamine (6-OHDA)-induced parkinsonism, Haghdoost-Yazdi and colleagues reported a dose-dependent reduction in the severity of motor symptoms following folic acid supplementation (Figure 5). Unexpectedly, in their study, HCY levels in rats supplemented with folic acid were not significantly reduced compared to the control group; in fact, the group receiving a high intake of folic acid had significantly higher HCY levels than the control group [317]. Moreover, in a Drosophila PD model expressing the new recessive allele of *parkin* (known to cause early-onset PD), parkc00062, folic acid protected against pupal lethality, high mortality, locomotor defects, elevated oxidative stress, and reduced metabolically active cellular status. Additionally, folic acid also improved mitochondrial respiration and increased ATP levels. These results support the potential of folic acid to alleviate challenges associated with parkin dysfunction in the Drosophila model and encourage further exploration of its role as a potential therapeutic strategy for parkin-mediated neurodegenerative diseases [318].

Amyotrophic lateral sclerosis (ALS) is another neurodegenerative disease associated with elevated HCY levels (Figure 5). A recent meta-analysis revealed that the cerebrospinal fluid HCY levels among ALS patients were significantly higher than those in controls, while no significant differences were observed in the plasma levels of HCY, folic acid, or vitamin B12 between ALS patients and controls [319]. However, some studies have reported lower folate levels in ALS patients compared with healthy controls [320,321,322] (Figure 5). In SOD1G93A transgenic mice (a model of ALS), Zhang et al. observed a significant reduction in folic acid levels during the middle to late stages of the disease. Administering folic acid and vitamin B12 significantly delayed disease onset and extended the lifespan of ALS mouse models by lowering plasma HCY levels, preventing microglia and astrocyte activation, and inhibiting the expression of inducible nitric oxide synthase and TNF-α [323] (Figure 5).

Despite some controversy, folic acid supplementation appears to play a role in normalizing high HCY levels, which are often associated with conditions such as AD, PD, and ALS. While more extensive clinical trials are necessary to fully understand the long-term benefits and optimal dosing strategies, current evidence suggests that folic acid supplementation could become a valuable component of treatment protocols for neurodegenerative diseases.

### 6.4. The Role of Combined Supplementation with Folic Acid and Other B Vitamins in Neurodegeneration

Supplementation with folic acid and other B vitamins (vitamin B12 and vitamin B6) has also been considered to improve symptoms in patients with neurodegenerative diseases [25,317,324,325,326,327,328,329,330]. Chen et al. explored the impact of folic acid and vitamin B12 supplementation on inflammation and cognitive health in stable AD patients with a Montreal cognitive assessment score (MoCA) below 22 (normal is over 26), treated with the acetylcholinesterase inhibitor or memantine. This supplementation (1.2 mg/day folic acid and 50 µg/day vitamin B12) over 6 months significantly improved MoCA scores, including naming, orientation, and attention on the AD assessment scale-cognitive subscale (ADAS-Cog). Blood levels of folate, vitamin B12, and SAM increased, while HCY, SAH, and TNFα decreased [325]. In middle-aged and elderly individuals with hyperhomocysteinemia, B vitamin supplementation (800 µg/day folate, 10 mg/day vitamin B6, and 25 µg/day vitamin B12) significantly improved cognitive function. This suggests that B vitamin supplementation may provide therapeutic benefits in reducing cognitive decline related to high HCY levels [326].

On the opposite side, Boston et al. reported no significant differences in folate and vitamin B12 levels between AD cases and healthy controls. In addition, baseline levels of these vitamins did not predict disease progression in AD patients, contradicting previous findings suggesting a protective effect for folate and vitamin B12 on cognitive decline [331]. Similarly, Aisen et al. reported that a high-dose B vitamin regimen (5 mg folate, 25 mg vitamin B6, 1 mg vitamin B12) over 18 months did not slow down cognitive decline in individuals with mild to moderate AD. In contrast, this regime of supplementation was even associated with a higher incidence of depression-related adverse outcomes [332]. Another study still reported no significant decrease in the risk of cognitive impairment or dementia in men aged 75 years and older under daily supplementation with 2 mg folic acid, 400 µg vitamin B12, and 25 mg vitamin B6 [333].

Other studies suggest that improved nutrition can positively impact synaptic function and delay AD progression [334]. Nutrients like omega-3 fatty acids, docosahexaenoic acid (DHA), uridine, choline, folate, and vitamins B12, B6, E, and C are essential for neuronal synthesis, and deficiencies in these nutrients are common in AD patients [335,336]. Based on this, the medical food Souvenaid was designed to enhance synaptic membrane formation and function in AD patients. It contains a specific blend of nutrients including DHA (1200 mg), EPA (300 mg), UMP (625 mg), choline (400 mg), folic acid (400 µg), vitamin B6 (1 mg), vitamin B12 (3 µg), vitamin C (80 mg), vitamin E (40 mg), selenium (60 µg), and phospholipids (106 mg) [337,338]. Clinical trials demonstrated that Souvenaid improved memory performance in AD patients and maintained brain network organization, potentially countering AD-related network disruption [339,340,341,342]. However, as an add-on treatment, Souvenaid did not reduce cognitive decline in mild-to-moderate AD patients receiving standard care [343]. A cognitive improvement was also revealed in AD patients supplemented with another nutraceutical formulation (folate, alpha-tocopherol, vitamin B12, SAM, *N*-acetylcysteine, and acetyl-L-carnitine) over 3 months [344].

These findings demonstrate that formulations containing folate can enhance synaptic function in AD, thereby promoting improved cognition. However, despite this interesting evidence, it becomes harder to link the delay in AD pathology to folic acid.

### 6.5. Folate Supplementation Can Reduce Heart Damage Induced by Alzheimer’s Disease

Neuropathological and epidemiological studies have demonstrated a link between AD and several cardiovascular risk factors, emphasizing the need for strategies to reduce cardiovascular complications in this neurodegenerative disease [345]. In a triple-transgenic AD mice model, a high dose of folic acid (3 mg/dL folate in drinking water, combined with intragastric administration of 1.2 mg/kg folate every day) provided cardioprotection, as it increased SIRT1-related protein (AMPK, SIRT1, SOD-2) and survival signaling protein (IGF1 receptor, pPI3K, pAKT, pBad) levels and reduced apoptotic cardiac cells. This suggests a potential therapeutic benefit of folic acid for cardiac dysfunction in AD models [345].

Another study demonstrated that administering folic acid and folinic acid (12 mg/kg daily) to late-stage triple-transgenic AD mice reduced apoptotic cell numbers and apoptosis-related protein expression. The treatment activated the IGF1R/PI3K/AKT and SIRT1/AMPK pathways, resulting in increased SIRT1 expression, translating into reduced cardiac damage and enhanced survival. These findings suggest folic acid and folinic acid promote cell survival in aged triple-transgenic AD mice [346].

### 6.6. Pharmacological Agents Targeting Folate Metabolism in Neurodegeneration

The development of pharmacological agents targeting folate metabolism is seen as an innovative therapeutic strategy in neurodegeneration. Additionally, the development of drugs targeting folate, its precursors, or its products may yield agents capable of decreasing HCY levels and oxidative stress more effectively than folate alone [347].

Folate supplementation may not achieve the expected effectiveness, in part because of feedback mechanisms within the methionine cycle. SAH hydrolase, an enzyme that normally hydrolyzes SAH into HCY, can reverse its action in the presence of high HCY levels consequent to folate deficiency, resulting in SAH accumulation. Moreover, SAH, a byproduct of SAM transmethylation, acts as a potent competitive inhibitor of SAM-mediated methylation reactions. To lower HCY levels effectively, developing selective inhibitors of SAM and SAH could be a promising approach [347].

Increasing the bioavailability of folate and its ability to cross the blood–brain barrier (BBB) could enhance its effectiveness in preventing and treating neurodegenerative diseases. The most direct and common way to increase folate bioavailability in the brain is through dietary supplementation with folic acid or reduced forms of folate. However, developing folate analogs that can be more efficiently transported across the BBB and metabolized in the brain may improve the efficacy of folate in brain metabolic processes.

In recent years, there has been increasing interest in developing agents targeting the enzyme MTHFR, which plays a critical role in folate metabolism. This interest is supported by the fact that genetic variations in MTHFR can lead to impaired folate metabolism and are associated with certain neurodegenerative diseases [348]. Drugs that modulate MTHFR activity could potentially correct or compensate for these deficiencies. Additionally, modulating folate transporters through agents that can increase the activity or expression of these transporters at the BBB may be an effective strategy to facilitate folate uptake into the brain. Developing controlled-release formulations of folate can also help maintain stable and therapeutic levels of folate in the brain over time.

All these pharmacological strategies focusing on folate metabolism pathways may offer innovative approaches in neurodegeneration to improve brain health and potentially slow the progression of these pathologies.

## 7. Concluding Remarks

Folate-dependent 1C metabolism represents a complex metabolic pathway responsible for fundamental cellular processes that support the overall cellular homeostasis and, thereby, the function of living organisms. Since eukaryotic cells, including human cells, lack the capability to synthesize folate, it is crucial to obtain this essential vitamin through diet or supplementation. However, the bioavailability of folate is influenced by several factors, including its chemical form, the food matrix, cooking methods, genetic factors, and interactions with other nutrients and medications. The importance of adequate folate intake is particularly evident during periods of rapid growth and development, such as pregnancy and infancy, where folate requirements are significantly increased.

Biological processes dependent on folate include cell division, neural tube formation, neuronal function, and neurotransmitter synthesis. To secure this multitude of roles, folate metabolism requires multifaceted regulation involving transcriptional, translational, and post-translational mechanisms that ensure the proper function and balance of key enzymes. Genetic polymorphisms can significantly affect enzyme activity and folate pathway efficiency, while epigenetic modifications can alter gene expression patterns in response to folate availability. Understanding these regulatory mechanisms is crucial for comprehending how folate metabolism impacts health and how its dysfunction results in disease.

Situations characterized by perturbed folate metabolism likely facilitate cellular dysfunction and represent a trigger for the development of human diseases, such as cancer, and neuropsychiatric and neurodegenerative diseases. In cancer, folate metabolism’s role is dual-faceted. While folate is necessary for DNA synthesis and repair, facilitating normal cell growth, its dysregulation can lead to genomic instability and cancer progression. This understanding has led to the development of antifolate drugs and targeted therapies for FOLRs, which have become fundamental therapeutic alternatives in oncology. However, the relationship between folate and cancer is complex, with folate intake showing both protective and adverse effects depending on the cancer type and individual genetic factors.

On the other hand, current evidence suggests that with appropriate administration, folic acid supplementation could become a valuable component in the management of neurodegenerative diseases, potentially improving cognitive health and quality of life for affected individuals. Mechanistically, folic acid supplementation helps in mitigating the elevated HCY levels characteristic of several neurodegenerative diseases by accelerating the remethylation pathway (dependent on vitamins B12 and folate). Despite the promising evidence, the effectiveness of folic acid supplementation in neurodegeneration remains controversial, necessitating more extensive clinical trials to fully understand its benefits and improve treatment protocols. At this level, pharmacological approaches targeting folate metabolism, such as developing folate analogs with improved bioavailability and agents modulating MTHFR activity, may offer innovative therapeutic strategies in neurodegeneration. These approaches aim to lower HCY levels and oxidative stress more effectively than folate alone, potentially providing a more robust intervention for neurodegenerative diseases.

In summary, proper folate metabolism is indispensable for normal development and function of the nervous system, as well as for maintaining genomic stability and controlling cell proliferation, thus preventing cancer development and growth. Insights into its mechanisms have not only increased our understanding of congenital malformations, CNS conditions, and cancer but have also revealed promising alternatives for targeted interventions that could improve health outcomes.

## 8. Future Perspectives and Current Therapeutic Challenges

Given its importance, folate-based drugs have become a significant area of interest, particularly in the treatment of cancer. However, designing effective folate-based drug medications presents a myriad of challenges and requires careful consideration across various scientific and clinical domains.

### 8.1. Resistance Mechanisms to Antifolate Compounds

It is increasingly recognized that cells can acquire resistance to antifolates. These mechanisms may include impaired intracellular influx in these drugs, alterations in their intracellular targets, reduced intracellular polyglutamylation, or, characteristically, the involvement of efflux mechanisms mediated by P-glycoprotein (P-gp).

The reduced folate carrier (RFC) is a transmembrane protein responsible for transporting reduced folates (such as THF) into cells, thereby maintaining their optimal intracellular levels [349]. Alternatively, folates can enter cells via FOLRs or proton-coupled folate transporter (PCFT) [350]. It has been shown that RFC is the primary mechanism for cellular entry of classical antifolates. The uptake of propargyl-linked antifolates (such as CB3717 and ICI-198) and the purine synthesis inhibitor, 5,10-dideazatetrahydrofolate, is predominantly mediated by FOLRs [63]. In addition, pemetrexed is equally transported by PCFT and RFC at physiological pH [351]. Thus, alterations in the expression and/or function of these transporters may impact the intracellular concentration of antifolate drugs and, consequently, their therapeutic efficacy.

The altered expression and function of transcription factors resulting in transcriptional silencing of the hRFC promoter in leukemia cells have been linked to methotrexate resistance [352]. Methylation of the hRFC-B promoter has been linked to reduced rates of complete remission in primary CNS lymphomas treated with methotrexate [353]. Phosphorylation of tyrosine residues on heat shock cognate protein (HSC70) also controls the transport of methotrexate into cells through the HSC70-RFC system, thereby contributing to methotrexate resistance in lymphocytic leukemia cells [354]. Human leukemia cells resistant to the potent DHFR inhibitor, PT523, show a strong reduction (65–99%) in RFC gene expression and a cluster of four nearly consecutive mutations residing on a single RFC allele including L143P, A147V, R148G, and Q150Stop [355]. This supports the involvement of transcriptional silencing mechanisms and inactivating mutations in resistance to antifolate drugs.

Considering the limitations we face with antifolates transported by RFC, and noting that PCFT becomes more active in the acidic environments characteristic of tumors, much effort has been directed toward the rational design of novel 6-substituted pyrrolopyrimidine molecules that are selectively transported by PCFT. These compounds may show clinical relevance against malignant mesothelioma and non-small cell lung cancer [356]. Therefore, more efforts are needed to further advance the development of new candidate drugs in this category.

The dysfunction of the ATP-dependent cytoplasmic and mitochondrial enzyme folylpoly-γ-glutamate synthetase results in a significant decrease in intracellular levels of methotrexate polyglutamate and prevents their intracellular accumulation [357]. The activity of mitochondrial NADH dehydrogenase 1 beta subcomplex subunit 7 (also known as SQM1), which transfers electrons from NADH to the respiratory chain, also influences the transport of methotrexate. While SQM1 is reduced in human squamous carcinoma cells resistant to methotrexate, cell treatment with SQM1-liposomes restored methotrexate intracellular levels and cytotoxic effects [358].

Alterations in intracellular targets of antifolate drugs also contribute to cancer resistance to these agents. In osteosarcoma cells, DHFR overexpression is closely associated with methotrexate resistance [359]. Overexpression of DHFR also translates into poor survival in acute lymphoblastic leukemia patients [360,361]. Therefore, the development of novel antifolate drugs that target intracellular mechanisms other than DHFR (preferentially with no fluctuations between normal and cancer cells) may be beneficial for improving treatments.

### 8.2. Side Effects of Folate-Targeted Therapies

The therapeutic window of antifolate drugs is frequently restricted, meaning there is a fine line between therapeutic efficacy and toxicity. Methotrexate, for instance, can cause severe side effects such as bone marrow suppression, hematological toxicity, and nephrotoxicity [362].

It is widely recognized that variations in the RFC gene, as well as genes that regulate intracellular folate and antifolate processing, may contribute to antifolate toxicity by causing excessive accumulation within cells [63]. For instance, the MTHFR gene variant *C677T* is linked to hematological adverse reactions to pemetrexed in Chinese patients with non-small cell lung cancer [363]. Polymorphisms in the 5-aminoimidazole-4-carboxamide-ribonucleotide formyltransferase (*T102C*) and gamma-glutamyl hydrolase (*G91T*) genes have also been linked to both adverse reactions and therapeutic outcomes [363]. Therefore, mapping these genetic variations may assist in predicting treatment outcomes.

Another important aspect related to therapy with antifolate drugs is nephrotoxicity, particularly with methotrexate. Currently, this can be prevented by administering folinic acid, which stimulates the tubular secretion of methotrexate via organic anion transporter K1 (OAT-K1) and OAT-K2. Changes in the expression and/or activity of these transporters may pose adverse toxic effects to kidneys in methotrexate-treated patients because of longer methotrexate exposure [364]. In addition, signs of pulmonary toxicity, including coughing, wheezing, and breathlessness, are manifested in patients under methotrexate treatment [365]. The development of novel drugs that target the folate metabolism, but are devoid of kidney and pulmonary toxicity, is of clear importance.

On the other hand, compounds that inhibit P-gp and/or multidrug resistance proteins (MRPs), including verapamil and quinidine, and the naturally occurring inhibitors flavonoids, coumarins, terpenoids, alkaloids, and saponins, among others, can promote the intracellular accumulation of methotrexate, thereby increasing the risk of toxicity [366,367].

Overall, designing drugs with improved safety profiles and developing strategies for the long-term management of side effects are critical considerations.

### 8.3. Targeted Therapies and Delivery Systems

The route of administration significantly impacts the pharmacokinetics and pharmacodynamics of any drug. Oral administration is convenient but may be limited by poor bioavailability and variability in absorption. Parenteral routes provide more consistent drug levels but are less convenient. To bypass these problematics, investigations in innovative delivery systems, including nanoparticles and liposomes, have been explored to improve drug delivery and efficacy.

As mentioned above, one of the most promising strategies in targeted antifolate therapy involves leveraging the overexpression of FOLRs on cancer cells [368]. Folate receptor-targeted therapies use folate or folate analogs conjugated to antifolate drugs, allowing for selective delivery to cancer cells with minimal impact on normal tissues [368]. Folate-drug conjugates, such as vintafolide, have shown promise in clinical trials for ovarian and lung cancers [218].

On the other hand, nanotechnology offers modern solutions for enhancing the specificity and efficacy of drugs, including antifolate molecules. Liposomal methotrexate formulations have shown improved pharmacokinetics and reduced toxicity in preclinical studies [369]. Recently, Lodhi and colleagues demonstrated that engineered magnetic nanoparticles with methotrexate and anti-FR antibody (MNP-MTX-FR) showed strong cytotoxic effects, indicating that the antibody-coated drug has significant potential to effectively target and kill cancer cells in the drug delivery process [370]. Thus, targeted therapies and delivery systems can enhance delivery to cancer cells, reducing off-target effects and improving therapeutic results. Continuous investigation and development in this area are likely to provide significant advancements in the near future.

In the context of neurodegenerative and other brain diseases, the most challenging aspect in developing new drugs is the BBB [371]. Although selective permeability represents the most significant challenge, the BBB is also equipped with efflux transporters, such as P-gp and MRPs, that actively eliminate drugs from the brain. These transporters reduce the intracerebral concentration of many pharmacological agents, diminishing their therapeutic efficacy. Additionally, we must consider that physiological factors, including enzymatic degradation of drugs within brain endothelial cells, complicate drug delivery. To bypass these limitations, the following strategies can be explored: introducing chemical modifications to drugs, using direct delivery methods, utilizing nanotechnology-based approaches, or implementing biological strategies. Direct delivery methods, such as intracerebral injection or infusion, bypass the BBB entirely by delivering drugs directly into the brain tissue. However, these techniques are invasive and may face limited therapeutic acceptance. Techniques like convection-enhanced delivery (CED) use a pressure gradient to distribute therapeutic agents over a larger area of the brain [372]. Alternatively, intranasal delivery systems take advantage of the olfactory and trigeminal nerve pathways to transport drugs directly to the brain. This non-invasive method bypasses the BBB and has shown promise in delivering peptides, proteins, and small molecules to the CNS [373].

Modifying drugs to increase their lipophilicity represents another strategy for enhancing their ability to cross the BBB. However, this approach must balance increased lipophilicity with maintaining the drug’s pharmacological activity and reducing potential toxicity. Alternatively, the development of prodrugs that are bioactivated inside the brain can improve their BBB permeabilities and enhance their therapeutic effects once inside the CNS [374,375].

Nanoparticles can be engineered to carry drugs across the BBB. These tiny carriers can be designed to enhance drug stability, prolong circulation time, and facilitate targeted delivery to the brain [376]. Examples include liposomes, dendrimers, and polymeric nanoparticles. Particularly, functionalizing liposomes with ligands or antibodies that target BBB receptors can improve their uptake by brain endothelial cells [377].

Among the biological strategies that can be used to bypass the BBB, disruption techniques and receptor-mediated transport have been increasingly investigated. Temporary disruption of the BBB using methods such as focused ultrasound (FUS) combined with microbubbles or osmotic agents can increase BBB permeability and allow drug transport into the brain [378]. These techniques must be carefully controlled to avoid damage to the BBB and surrounding tissues. Developing receptor-mediated transport systems, such as transferrin receptors or insulin receptors, enables the transport of drugs conjugated to ligands that naturally cross the BBB. This strategy can facilitate the delivery of larger molecules, including proteins and antibodies, into the CNS [379].

## Figures and Tables

**Figure 1 ijms-25-09339-f001:**
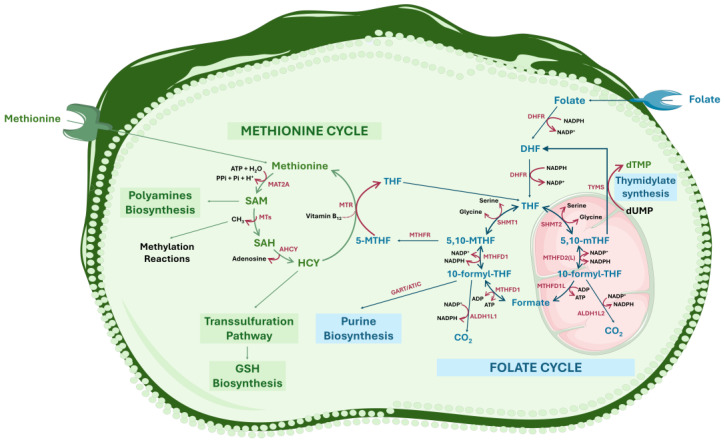
Folate-dependent one-carbon metabolism. Folate-dependent 1C metabolism includes the interconnected folate and methionine cycles, which are crucial for the de novo synthesis of purines and thymidylate, the biosynthesis of polyamines, and the production of glutathione via the transsulfuration pathway. Elevated levels of homocysteine (HCY) from the methionine cycle represent a risk factor for cardiovascular diseases, congenital anomalies such as neural tube defects, neurodegenerative disorders, and certain types of cancer. **ATIC**, 5-aminoimidazole-4-carboxamide ribonucleotide formyltransferase; **GART**, trifunctional polypeptide with phosphoribosylglycinamide formyltransferase, phosphoribosylglycinamide synthetase, phosphoribosylaminoimidazole synthetase activity; **Pi**, inorganic phosphate. The illustration was prepared using Microsoft PowerPoint 2021 software and Servier Medical Art images.

**Figure 2 ijms-25-09339-f002:**
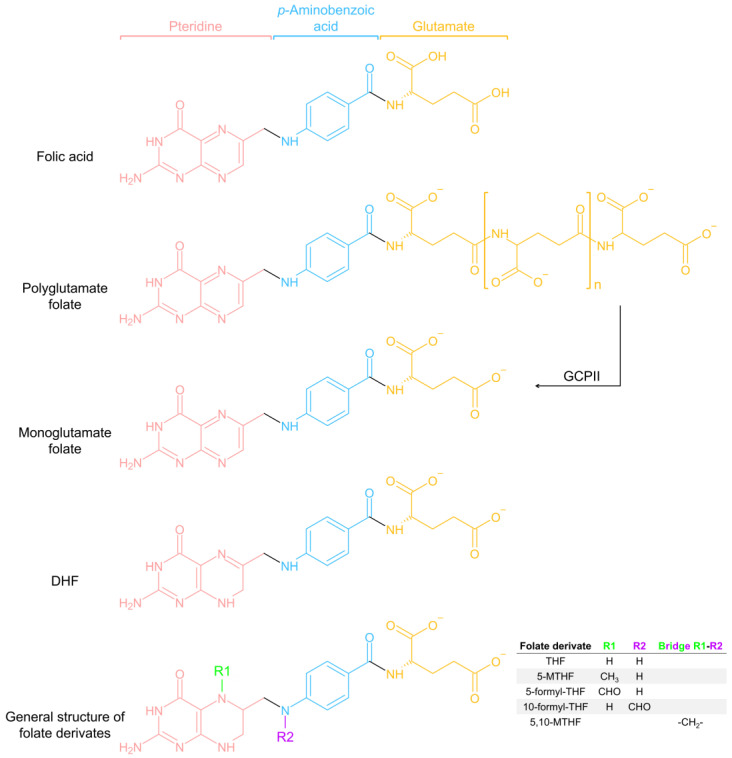
Chemical structures of folic acid, folate (in both polyglutamate and monoglutamate forms), and various folate derivatives. The chemical structures were drawn using ChemSketch freeware 2023.2.4 [32].

**Figure 3 ijms-25-09339-f003:**
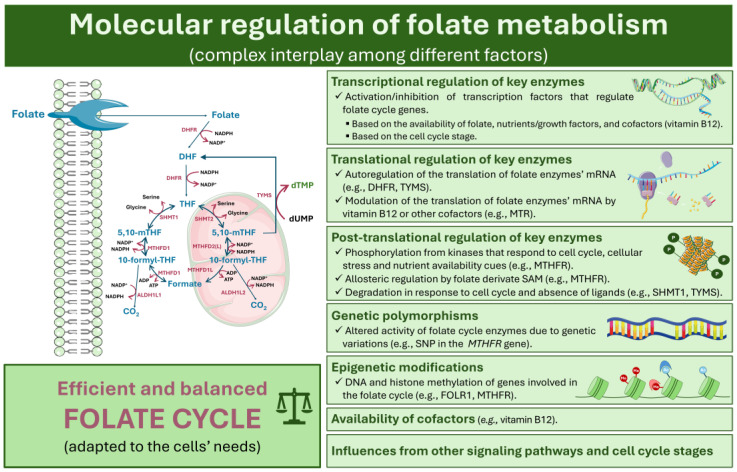
Molecular regulation of folate metabolism. Folate metabolism is regulated at different levels, including transcriptional, translational, and post-translational modifications, genetic polymorphisms, epigenetic modifications, availability of cofactors, and influences from other signaling pathways and cell cycle stages. The illustration was prepared using Microsoft PowerPoint 2021 software and Servier Medical Art images.

**Figure 4 ijms-25-09339-f004:**
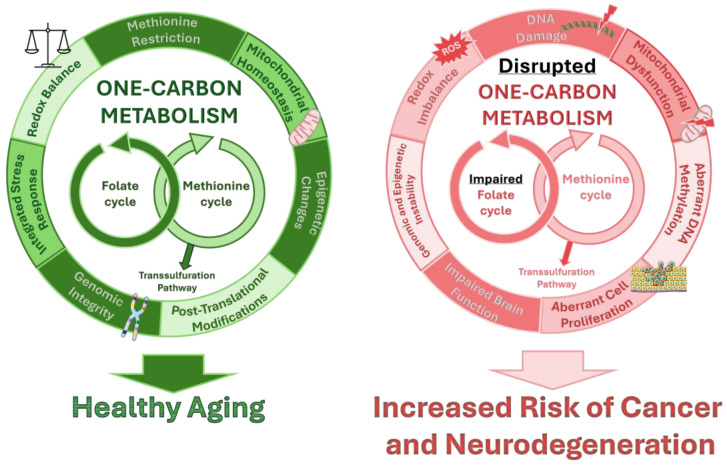
Involvement of folate-dependent one-carbon metabolism in vital physiological processes in organisms. Proper functioning of folate-dependent 1C metabolism is crucial for maintaining mitochondrial homeostasis, genetic integrity, precise epigenetic control, post-translational modifications, and redox balance. Disruption of its normal function can lead to DNA damage, abnormal DNA methylation, uncontrolled cell proliferation, genetic and epigenetic instability, mitochondrial dysfunction, and redox imbalance, thereby increasing the risk of cancer and neurodegeneration. Perturbations in folate-dependent 1C metabolism during pregnancy may result in congenital malformations and birth defects. The illustration was prepared using Microsoft PowerPoint 2021 software and Servier Medical Art images.

**Figure 5 ijms-25-09339-f005:**
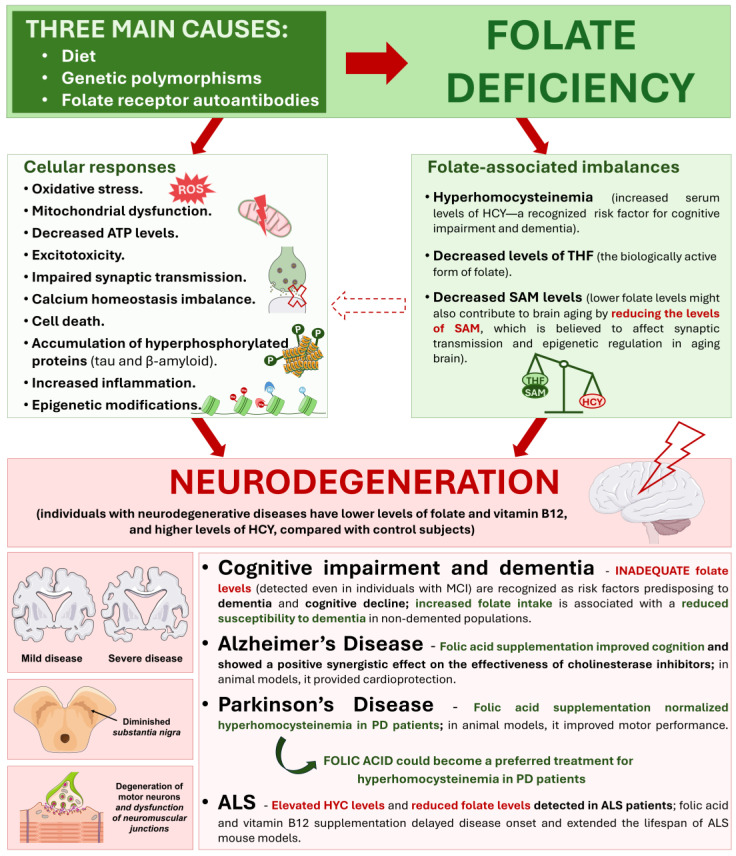
Mechanisms of neurodegeneration associated with folate deficiency and the role of folate supplementation in counteracting the neurodegenerative process. The illustration was prepared using Microsoft PowerPoint 2021 software and Servier Medical Art images.

**Table 3 ijms-25-09339-t003:** Detailed summary of preclinical and clinical studies on antifolate drugs, including descriptions, clinical applications, and observations.

Antifolate Drug	Description	Clinical Applications	Other Considerations	References
**Vintafolide (EC145)**	Folic acid conjugate with vinca alkaloid	Solid tumors (ovarian and non-small cell lung cancer)	- Showed promise for platinum-resistant ovarian cancer in a phase II clinical study (140 patients), but the phase III trial was suspended. - Clinical studies for lung cancer (43 and 203 patients) showed a tendency for the greatest benefit in patients with 100% of lesions positive for FOLR.	[229,230,231,232]
**Pafolacianine (OTL38)**	Conjugate between folate and NIR fluorescent dye	Lung cancer	- Clinical study (411 patients tested). - Dose of 0.025 mg/kg, 3 to 24 h before surgery. - Undergoing clinical trials.	[220]
**EC2629**	Folate conjugate of a DNA crosslinking agent	In combination with FOLR-positive KB human xenografts in mice	- Preclinical study. - Mice (five animals)—0.3 µmol/kg once per week over two weeks. - Rat (three animals)—0.15 µmol/kg once per week over two weeks. - Demonstrated 100% cure rates at very low doses.	[204]
**Farletuzumab (MORAb003)**	Monoclonal antibody therapy	Solid tumors	- Phase III clinical study on ovarian cancer (1100 patients tested). - Dose of 1.25 mg/kg or 2.5 mg/kg farletuzumab plus taxane/CBDCA against placebo plus taxane/CBDCA; weekly administration. - Improvement in progression-free survival was not statistically significant. - Phase II clinical study on lung cancer (130 patients). - Dose of 7.5 mg/kg triweekly in combination with chemotherapy.	[233,234]
**MORAb202**	ADC with FOLR binding antibody	Advanced solid tumors	- Clinical study (22 patients). - Dose of 0.3 to 1.2 mg/kg every 3 weeks. - Length of the study—up to ~8 months. - Well-tolerated in phase I study.	[222]
**Mirvetuximab soravtansine (IMGB853) or Elahere™**	ADC targeting FOLR1	FOLRα-positive ovarian, fallopian tube, and peritoneal cancers	- Clinical studies (106 and 366 patients). - Approved in the USA in 2022. - Dose of 6 mg/kg administered intravenously every 3 weeks per cycle. - Tumor regression consistent in several clinical trials.	[235,236]
**CAR T-cell therapies**	Cellular immunotherapy	Ovarian cancer	- Preclinical study (high-grade serous ovarian cancer patient samples). - Promoted the lysis of patient-derived tumor cells after 16 h of co-culture. - In NOD SCID gamma (NSG; NOD.CgPrkdcscidIl2rgtm1Wjl/SzJ) mice (1 × 10^7^ CAR T cells, i.v.), CAR T-cell were well-tolerated and reduced tumor burden (after 21 days).	[225]
**5-MTHF**	Albumin-binding radioconjugates	Cervical carcinoma	- Preclinical study. - In athymic nude mice (CD-1 Foxn-1/nu; 6–9 animals/group) [^177^Lu]Lu-6R-RedFol-1 or [^177^Lu]Lu-OxFol-1 (5 nmol) demonstrated slower tumor growth and increased median survival time (measured up to 56 days after injection).	[228]

**ADC**, antibody–drug conjugates; **CAR T**, engineered patients’ immune cells, **DNA**, deoxyribonucleic acid; **FOLR**, folate receptor; **NIR**, near-infrared spectroscopy; **5-MTHF**, 5-methyltetrahydrofolate.

## Data Availability

Not applicable.

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
