# Peer review of "Unveiling the Therapeutic Potential of Folate-Dependent One-Carbon Metabolism in Cancer and Neurodegeneration"

_ijms, 2024, doi:10.3390/ijms25179339_

Round 1

Reviewer 1 Report

Comments and Suggestions for Authors

The manuscript provides valuable insights into the therapeutic potential of targeting folate-dependent one-carbon metabolism in both cancer and neurodegenerative diseases. It covers the mechanisms by which this metabolic pathway influences disease progression and highlights potential strategies for therapeutic intervention.

1-    I recommend creating an illustration to effectively conclude the section on the molecular regulation of folate metabolism.

2-    If applicable, please revise the text to include a citation for the software used to generate the illustrations.

3-    ALDH1L1, an enzyme involved in folate metabolism that catalyzes the conversion of 10-formyl-THF to THF, has been identified in several studies as a potential tumor suppressor. Please include a discussion of this finding in the paper under the section titled "Folate Metabolism and Cancer."

4-    Other folate drugs, including Pralatrexate, Raltitrexed, Trimethoprim, and Sulfamethoxazole, should be discussed and added to Table 1.

Author Response

REVIEWER 1

The manuscript provides valuable insights into the therapeutic potential of targeting folate-dependent one-carbon metabolism in both cancer and neurodegenerative diseases. It covers the mechanisms by which this metabolic pathway influences disease progression and highlights potential strategies for therapeutic intervention.

Authors’ response: We greatly acknowledge the Reviewer for the positive feedback on the manuscript.

1 - I recommend creating an illustration to effectively conclude the section on the molecular regulation of folate metabolism.

Authors’ response: We greatly appreciate the Reviewer's suggestion. In line with this recommendation, we have created a new figure to summarize the information contained in the section on the molecular regulation of folate metabolism (Figure 3 in the revised manuscript). We hope the new figure meets the Reviewer's expectations.

2 - If applicable, please revise the text to include a citation for the software used to generate the illustrations.

Authors’ response: We greatly appreciate the Reviewer's suggestion. Figures 1, 3, 4, and 5 were prepared using Microsoft PowerPoint software and Servier Medical Art images. The chemical structures shown in Figure 2 were drawn using ChemSketch freeware. In the revised manuscript, we have included this information in the figure legends, and an acknowledgment to Servier Medical Art by Servier has also been included. We hope this clarification meets the Reviewer's expectations.

3 - ALDH1L1, an enzyme involved in folate metabolism that catalyzes the conversion of 10-formyl-THF to THF, has been identified in several studies as a potential tumor suppressor. Please include a discussion of this finding in the paper under the section titled "Folate Metabolism and Cancer."

Authors’ response: We greatly appreciate the Reviewer's suggestion. In line with this suggestion, we have provided information on the potential tumor suppressor activity of ALDH1L1 and how cancer cells downregulate the expression of this enzyme as an adaptive response.

In the revised manuscript, this information reads as follows:

“Additionally, several studies have identified ALDH1L1 as a potential tumor sup-pressor [190-192]. ALDH1L1 may inhibit tumor growth by reducing the availability of 10-formyl-THF, a critical component for purine synthesis, thereby limiting DNA synthesis and cell proliferation. Thus, as an adaptive response, most cancer cells show a downregulation or loss of ALDH1L1 [193, 194], which results in an increased pool of 10-formyl-THF, supporting enhanced nucleotide biosynthesis and tumor progression”.

We hope these alterations meet the Reviewer's expectations.

4 - Other folate drugs, including Pralatrexate, Raltitrexed, Trimethoprim, and Sulfamethoxazole, should be discussed and added to Table 1.

Authors’ response: We greatly appreciate the Reviewer's suggestions. In line with these recommendations, we incorporated additional antifolate drugs, such as pralatrexate and raltitrexed, into subsection 5.2, “Pharmacological agents targeting folate metabolism in cancer”. Additionally, this information has been included in Table 2 (corresponding to Table 1 in the initial document).

In the revised manuscript, this information reads as follows:

“Pralatrexate is another antifolate drug that inhibits the enzyme DHFR and is primarily used in relapsed or refractory peripheral T-cell lymphoma [196, 197]. Pemetrexed inhibits TYMS, THF reductase, and glycinamide ribonucleotide formyltransferase and has proven efficacy for non-squamous non-small cell lung cancer [198]. Raltitrexed works by inhibiting TYMS and is primarily used in advanced colorectal cancer [199]. 5-fluorouracil (5-FU) specifically targets TYMS and is commonly used as a primary chemotherapy option for colorectal cancer, typically resulting in response rates of around 60-65% [200, 201]. Inhibitors targeting purine or pyrimidine synthesis have also been utilized in hematological malignancies. For instance, 6-mercaptopurine (6-MP), a thio-substituted purine analogue that inhibits de novo purine synthesis, is frequently used in combination with methotrexate as maintenance therapy in childhood acute lymphoblastic leukemia [202]”.

Regarding trimethoprim and sulfamethoxazole, these are antifolate drugs used to treat bacterial infections, and their use in cancer treatment has not yet been established. Thus, considering that the scope of the manuscript is to explore the therapeutic potential of folate-dependent one-carbon metabolism in cancer and neurodegeneration, we believe that these two antifolates do not fit within the scope of the manuscript.

We hope these alterations meet the Reviewer's expectations.

Reviewer 2 Report

Comments and Suggestions for Authors

Dear Authors

i have read your review paper with the greatest pleasure. The undertaken topic is very important and the quality of the presentation of so many complex data is smooth, clear and logic.

I have only minor comments to your excellent work.

1. Please, add a small table summarizing the most rich sources of folic acid together with - either folic acid content or the percentage of the daily reccommentations in 100 g

2. please, show the chemical structure of folic acid and its derivatives

3.for a more easy reading, i would recommend to add some subsections to the section 5.2. and divide it to small subchapters but only if you find it reasonable

4. please, modify table 2 by: adding an additional column with the type of experiment - a clinical or preclinical; for the cell lines please give an exact name (symbols) of cell lines; please, indicate how large was the group of patientis tested and introduce the length of the study, eventually the doses, too.

5. please, prepare an additional figure that summarizes the effects of folic acid on the neurodegeneration proces to better underline the mechanisms  

Author Response

REVIEWER 2

I have read your review paper with the greatest pleasure. The undertaken topic is very important and the quality of the presentation of so many complex data is smooth, clear and logic.

Authors’ response: We greatly acknowledge the Reviewer for the positive feedback on the manuscript.

1. Please, add a small table summarizing the most rich sources of folic acid together with - either folic acid content or the percentage of the daily recommendations in 100 g.

Authors’ response: We greatly appreciate the Reviewer's suggestion. In line with this recommendation, we have included a new table (Table 1 in the revised manuscript), listing the richest natural food sources of folate and their nutrient content [the total amount of folate per 100 g of food and the percentage of the recommended daily intake (per 100 g)]. We hope these alterations meet the Reviewer's expectations.  

2. Please, show the chemical structure of folic acid and its derivatives.

Authors’ response: We greatly appreciate the Reviewer's suggestion. In line with this recommendation, we have prepared a figure depicting the structures of folic acid, folate (in both polyglutamate and monoglutamate forms), and various folate derivatives (Figure 2 in the revised manuscript). We hope this new figure meets the Reviewer's expectations.  

3. For a more easy reading, I would recommend to add some subsections to the section 5.2. and divide it to small subchapters but only if you find it reasonable.

Authors’ response: We greatly appreciate the Reviewer's suggestion. In line with a recommendation from Reviewer 3 (“Consider splitting this section and integrating its content into Sections 6 and 7, while reducing repetitive information”), we have integrated the content of Section 5.2 (in the original manuscript) into Section 6, “Folate metabolism and neurodegeneration” (in the revised manuscript). Thus, we hope these changes meet the suggestions of both Reviewers.  

4. Please, modify table 2 by: adding an additional column with the type of experiment - a clinical or preclinical; for the cell lines please give an exact name (symbols) of cell lines; please, indicate how large was the group of patients tested and introduce the length of the study, eventually the doses, too.

Authors’ response: We greatly appreciate the Reviewer's suggestions. In line with these recommendations, we have updated this table (Table 3 in the revised manuscript) by incorporating information regarding the type of study (clinical or preclinical), the number of patients tested, the length of the study, and the doses evaluated. We hope this new table meets the Reviewer’s expectations.

5. Please, prepare an additional figure that summarizes the effects of folic acid on the neurodegeneration process to better underline the mechanisms.

Authors’ response: We greatly appreciate the Reviewer's suggestion. In line with this recommendation, we have prepared a new figure summarizing the effects of folic acid on the neurodegeneration (Figure 5 in the revised manuscript). We hope this new figure meets the Reviewer's expectations.  

Reviewer 3 Report

Comments and Suggestions for Authors

This manuscript provides an extensive review of folate-mediated one-carbon metabolism in both cancer and neurodegeneration. Given the considerable amount of content, I suggest focusing on recently research topic to enhance clarity and depth. Below are my detailed comments: 

1.        Figure 1: The arrows in the figure are excessively large, particularly the dark blue arrow, which obscures the text between 5-10-MTHF and 10-formyl-THF. I recommend resizing the arrows to improve the figure's readability.

2.        Lines 102-108: The caption for Figure 1 is too lengthy. It would be more appropriate to move the detailed explanation to the main text. Additionally, if abbreviations are already defined in the main text, there is no need to repeat them in the caption.

3.        Line 109: There are two consecutive semicolons (;). Please remove one to correct this punctuation error.

4.        Section 5 Title and Content: Although the title mentions "cancer," the content does not adequately address the relevance of folate metabolism to cancer. Consider splitting this section and integrating its content into sections 6 and 7, while reducing repetitive information.

5.        Section 8 Title: The title of this section is confusing and contains unnecessary repetitive statements. I suggest revising it for clarity and conciseness or remove this part.

6.        The manuscript should include a conclusion section to summarize the key findings and propose future research directions.

7.        Literature Citations: The manuscript includes several outdated references. I recommend updating these to focus on recent studies that are more relevant to the current state of research.

Overall, while the manuscript offers valuable insights into folate-mediated one-carbon metabolism, improvements in organization and focus are needed.

Author Response

REVIEWER 3

This manuscript provides an extensive review of folate-mediated one-carbon metabolism in both cancer and neurodegeneration. Given the considerable amount of content, I suggest focusing on recently research topic to enhance clarity and depth. Below are my detailed comments:

1. Figure 1: The arrows in the figure are excessively large, particularly the dark blue arrow, which obscures the text between 5-10-MTHF and 10-formyl-THF. I recommend resizing the arrows to improve the figure's readability.

Authors’ response: We greatly appreciate the Reviewer's suggestion. In the revised manuscript, we have resized the arrows in Figure 1. We hope these alterations improve the figure's readability and meet the Reviewer's expectations.

2. Lines 102-108: The caption for Figure 1 is too lengthy. It would be more appropriate to move the detailed explanation to the main text. Additionally, if abbreviations are already defined in the main text, there is no need to repeat them in the caption.

Authors’ response: We greatly appreciate the Reviewer's comments and suggestions. In the revised manuscript, we have shortened the caption for Figure 1 by retaining some detailed information only in the main text and removing abbreviations that were already defined there.

In the revised manuscript, the legend of figure 1 reads as follows:

“Figure 1. Folate-dependent one-carbon metabolism.

Folate-dependent 1C metabolism includes the interconnected folate and methionine cycles, which are crucial for the de novo synthesis of purines and thymidylate, the biosynthesis of polyamines, and the production of glutathione via the transsulfuration pathway. Elevated levels of homocysteine (HCY) from the methionine cycle represent a risk factor for cardiovascular diseases, con-genital anomalies such as neural tube defects, neurodegenerative disorders, and certain types of cancer. ATIC, 5-aminoimidazole-4-carboxamide ribonucleotide formyltransferase; GART, tri-functional polypeptide with phosphoribosylglycinamide formyltransferase, phosphoribosyl-glycinamide synthetase, phosphoribosylaminoimidazole synthetase activity; Pi, inorganic phosphate. The illustration was prepared using Microsoft PowerPoint software and Servier Medical Art images”.

We hope these alterations meet the Reviewer's expectations.  

3. Line 109: There are two consecutive semicolons (;). Please remove one to correct this punctuation error.

Authors’ response: We greatly appreciate the Reviewer's suggestion. In the revised manuscript, a semicolon was removed at the indicated place.

4. Section 5 Title and Content: Although the title mentions "cancer," the content does not adequately address the relevance of folate metabolism to cancer. Consider splitting this section and integrating its content into sections 6 and 7, while reducing repetitive information.

Authors’ response: We greatly appreciate the Reviewer's suggestions. In the revised manuscript, we have removed Section 5, “Relevance of folate metabolism in cancer and neurodegeneration”, and integrated its content into other sections. The first part, “The importance of folate metabolism in cell division”, has been incorporated into Section 5, “Folate metabolism and cancer”. The second part, “Roles of folate metabolism in neural tube formation, neuronal function, and neurotransmitter synthesis”, is now included in Section 6, “Folate metabolism and neurodegeneration”. We hope these alterations meet the Reviewer’s suggestions.  

5. Section 8 Title: The title of this section is confusing and contains unnecessary repetitive statements. I suggest revising it for clarity and conciseness or remove this part.

Authors’ response: We greatly appreciate the Reviewer's suggestion. In the original document of the manuscript, Section 8 refers to “Concluding remarks”. Therefore, we believe that this commentary does not belong to that section but to the subsection titled “Role of the combination of folic acid and other B vitamins supplementation in neurodegeneration”, which was, in fact, confusing. Thus, to enhance clarity and conciseness, we have revised the title of this section to “The role of combined supplementation with folic acid and other B vitamins in neurodegeneration”. We hope this change meets the Reviewer’s suggestion. However, if that is not the case, we are still open to revising it according to the Reviewer’s expectations.

6. The manuscript should include a conclusion section to summarize the key findings and propose future research directions.

Authors’ response: We greatly appreciate the Reviewer's suggestions. The revised manuscript includes a conclusion section (Section 7: “Concluding remarks”) and a final section on future perspectives (Section 8: “Future perspectives and current therapeutic challenges'). We hope these sections meet the Reviewer’s expectations.

7. Literature Citations: The manuscript includes several outdated references. I recommend updating these to focus on recent studies that are more relevant to the current state of research.

Authors’ response: We greatly appreciate the Reviewer's comments. In the revised manuscript, we have updated some outdated references to more relevant and recent studies. We hope these changes meet the Reviewer’s expectations.

Round 2

Reviewer 3 Report

Comments and Suggestions for Authors

Nice review. All my comments have been resolved and I have no further concerns.